# Early onset diagnosis in Alzheimer's disease patients via amyloid-β oligomers-sensing probe in cerebrospinal fluid

Jusung An [1,11], Kyeonghwan Kim [2,3,11], Ho Jae Lim [4], Hye Yun Kim [2,3], Jinwoo Shin [1], InWook Park [2,3], Illhwan Cho [2,3], Hyeong Yun Kim [2], Sunghoon Kim [2,5,6], Catriona McLean [7], Kyu Yeong Choi [8], YoungSoo Kim [2,3] ✉, Kun Ho Lee [4,8,9] ✉ & Jong Seung Kim [1,10] ✉

Amyloid-β (Aβ) oligomers are implicated in the onset of Alzheimer's disease (AD). Herein, quinoline-derived half-curcumin-dioxaborine (Q-OB) fluorescent probe was designed for detecting Aβ oligomers by finely tailoring the hydrophobicity of the biannulate donor motifs in donor-π-acceptor structure. Q-OB shows a great sensing potency in dynamically monitoring oligomerization of Aβ during amyloid fibrillogenesis in vitro. In addition, we applied this strategy to fluorometrically analyze Aβ self-assembly kinetics in the cerebrospinal fluids (CSF) of AD patients. The fluorescence intensity of Q-OB in AD patients' CSF revealed a marked change of log ($I/I_O$) value of $0.34 \pm 0.13$ (cognitive normal), $0.15 \pm 0.12$ (mild cognitive impairment), and $0.14 \pm 0.10$ (AD dementia), guiding to distinguish a state of AD continuum for early diagnosis of AD. These studies demonstrate the potential of our approach can expand the currently available preclinical diagnostic platform for the early stages of AD, aiding in the disruption of pathological progression and the development of appropriate treatment strategies.

Alzheimer's disease (AD), the most common type of dementia, affects an increasingly high number of individuals worldwide[1,2]. According to the amyloid cascade hypothesis, amyloid-β (Aβ)-species (e.g., monomers, oligomers, fibrils, and plaques) are major hallmarks of AD from early onset[3–5]. Incipiently, amyloid precursor proteins are sequentially cleaved by β- and γ-secretase complex, and individually varied amounts of Aβ monomers are produced in cortex and hippocampus[5]. Aβ self-assembly leads to the formation of elongated fibrillar aggregates with highly ordered cross-β-sheet structures, insoluble Aβ fibrils.

A significant fraction of age-matched individuals with AD has Aβ deposits, but it is regarded as a minor cause of AD because it does not correlate well with the severity of cognitive impairment[6]. Instead, small aggregates-comprising aqueous soluble Aβ oligomers during the dynamic aggregation cascade are pathophysiologically responsible for the synaptotoxicity, inflammation, tau hyperphosphorylation, nonfibrotic tangling, and neuronal death in the brain, resulting in cognitive impairment[7–9]. Hence, Aβ oligomers-mediated pathology of AD is growing attention for developing future diagnostics and therapeutic

[1]Department of Chemistry, Korea University, Seoul 02841, Korea. [2]Department of Pharmacy, College of Pharmacy, Yonsei University, Incheon 21983, Korea. [3]Yonsei Institute of Pharmaceutical Sciences, College of Pharmacy, Yonsei University, Incheon 21983, Korea. [4]Department of Biomedical Science, Chosun University, Gwangju 61452, Korea. [5]Medicinal Bioconvergence Research Center, Institute for Artificial Intelligence and Biomedical Research, Gangnam Severance Hospital, Yonsei University, Incheon 21983, Korea. [6]College of Pharmacy, College of Medicine, Interdisciplinary Biomedical Center, Gangnam Severance Hospital, Yonsei University, Incheon 21983, Korea. [7]Department of Pathology, The Alfred Hospital, Melbourne 3004, Australia. [8]Gwangju Alzheimer's & Related Dementia Cohort Research Center, Chosun University, Gwangju 61452, Korea. [9]Department of Neural Development and Disease, Korea Brain Research Institute, Daegu 41062, Korea. [10]TheranoChem Incorporation, Seongbuk-gu, Seoul 02856, Korea. [11]These authors contributed equally: Jusung An, Kyeonghwan Kim. ✉e-mail: y.kim@yonsei.ac.kr; leekho@chosun.ac.kr; jongskim@korea.ac.kr

targets, as it is more closely correlated with the symptoms of dementia observed in AD than Aβ monomers or fibrillar deposits are[10,11]. Besides, given the long interval between the pathology resulting from amyloid fibrillogenesis and the onset of cognitive deficits, detecting Aβ oligomers can provide an excellent opportunity for early diagnosis of AD. Indeed, as various up-to-date AD treatment methods are emerging[12–14], the importance of early disease diagnosis is being emphasized to consider well-timed intervention of treatment, raising the possibility of curing cognitive impairment[15,16]. However, current clinical diagnosis methods, including cognitive assessment questionnaires, functional behavioral assessments, and positron emission tomography (PET), are still based on symptomatology and are accurate in only 63−90% of cases with dementia[17]. Additionally, these methods do not allow early-AD diagnosis and can only be applied when the disease has already progressed extensively. Thus, it requires a reliable early-diagnosing method in the current clinical area of AD.

Recently, in vitro diagnostic technologies toward potential bio-markers have been extensively investigated using antibody-based enzyme-linked immunosorbent assay (ELISA) for early diagnosis of AD[18–21]. Nevertheless, it has disadvantages of ELISA, such as the high-cost loss, complexity, and time-consuming procedure compared to a small molecule-based method. Hence, the development of small molecule probes has been paid increasing attention for application in diagnosis and drug discovery for AD, which may help overcome the limitations of conventional and ELISA-based AD diagnosis techniques. Small molecule-based fluorescence spectroscopy is one of the most practical and affordable technologies, typically non-invasive, convenient, cost-effective, and high-resolution in nature for conducting

basic research and preclinical studies on AD[22]. For use of fluorescence technology (e.g., optical imaging, quantification, and disease diagnosis), it is required to employ a suitable fluorescent probe, which has appropriate physicochemical/photophysical properties. Especially, the detection tools used in a biological environment, including body fluid samples, must have high sensitivity and selectivity so that they can be applied to specific misfolded amyloid proteins for examination of AD pathology[23]. Efficient detection of a misfolded amyloid proteins can be achieved in several ways, including functional selectivity, wherein the fluorescent behavior of the probe relies upon the milieu, binding affinity, or conformational changes[24,25]. However, fluorometric application for Aβ oligomers is technically challenging by the poor understanding of the oligomers' pathological and physicochemical properties, such as heterogeneous three-dimensional structure (polymorphic nature), unsustainability in physiological conditions, and the minute amount in biofluids[26]. Consequently, structural design strategy, by exploring the binding sites of an Aβ oligomer-specific sensor, is still shrouded in mystery to fulfill demanded specificity, selectivity, and sensitivity. Therefore, an exceptional strategy needs to be developed for discovering fluorophores that can directly trace the formation of Aβ oligomers in proteinaceous dynamic self-assembly.

Small molecule fluorescent dyes toward various states of Aβ oligomers have merely been reported for application in the following three categorized design strategies (Fig. 1 and Supplementary Table 1). (1) Coincidental discovery: At the initial stage of the current research area, the Aβ oligomer detection method was reported by utilizing the unexpected photophysical properties of fluorophores that could be used as a probe for detecting conformational changes in oligomeric

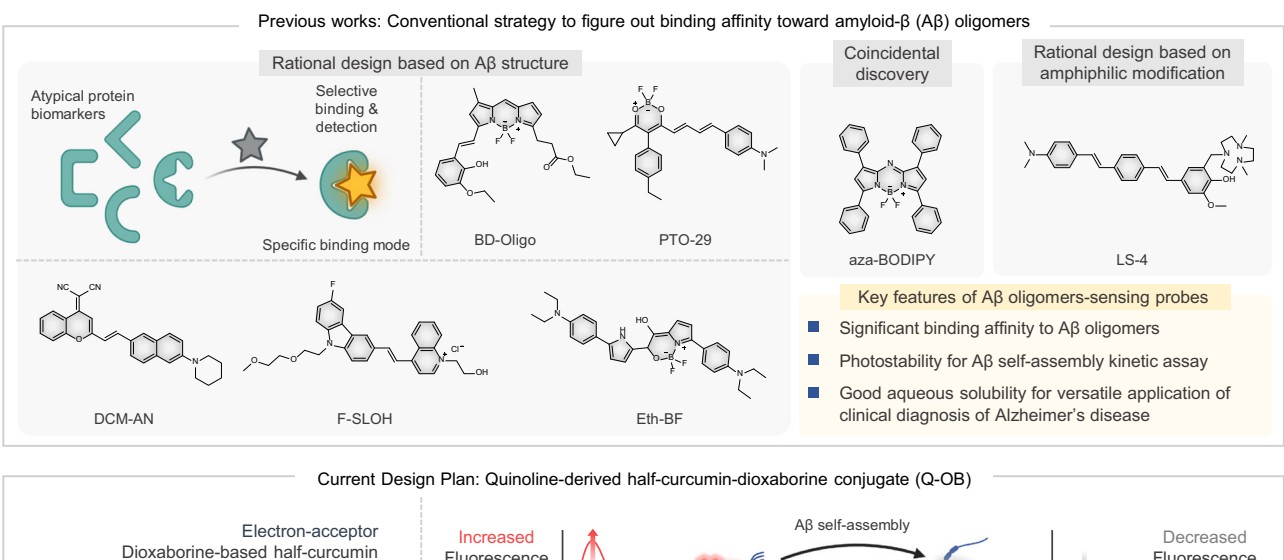

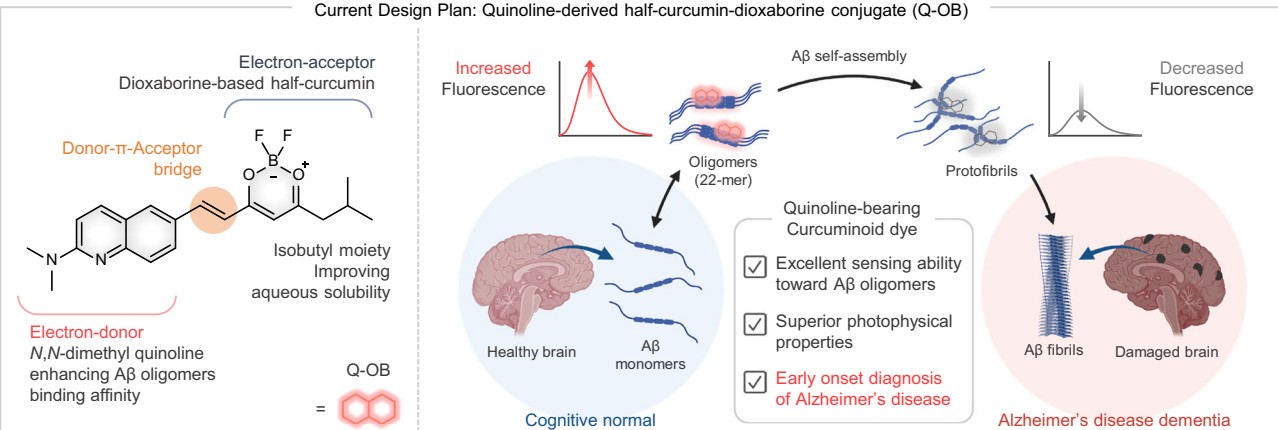

**Fig. 1 | Small molecule-based fluorescent sensor for amyloid-β (Aβ) oligomers; schematic illustration of hydrophobicity tailoring design strategy underlying the use of quinoline-derived half-curcumin-dioxaborine (Q-OB) as an Aβ oligomers-sensing probe.** Adapted from "Protein Aggregation Pathways," by BioRender.com (2023). Retrieved from https://app.biorender.com/biorender-templates.

$A\beta_{1-42}$[27]. The methods, however, used to design chemical structures in early research were not always rational, which caused their fluorescent response by Aβ oligomers. (2) Rational design based on Aβ oligomer structures: Some previous studies have used X-ray-determined Aβ trimers derived from Aβ peptides as a working model for toxic Aβ oligomers to design small molecule probes[28–32]. Computational methods also offer possible approximations as a logical starting point for molecule design. (3) Rational design based on amphiphilic modification: Recent research has revealed that amphiphilic molecules bearing hydrophilic fragments interacting with hydrophobic fibrillar fragments of Aβ could enhance binding affinity toward Aβ oligomers, to be utilized for in vivo imaging[33]. Conjugating hydrophilic and hydrophobic motifs remarkably enhances the binding affinity for Aβ. Indeed, these previous studies exhibited marked fluorescence response and good binding affinities for Aβ fibrils and oligomers. However, these approaches could not show a sufficient selective photo-response for Aβ oligomers in spontaneous Aβ assembly kinetics; they were not employed for determining and diagnosing the pathogenesis of AD as actual clinical biofluid samples.

In our previous study, a naphthalene-derived fluorophore, NAP-OB, could detect Aβ plaques with picomolar-scale binding affinity and sensitivity[25]. Using this backbone structure of NAP-OB as an excellent starting point, we herein employed hydrophobicity tailoring by replacing the naphthalene-based biannulated ring with quinoline, which may enhance the hydrogen-bonding interaction between engaged Aβ fragment and nitrogen atoms of quinoline[34–36]. We rationally combined the components of dioxaborine-based half-curcumin as an electron-acceptor moiety (A) and *N,N*-dimethylquinolin-2-amine as an electron-donor (D) to give D-π-A-type intramolecular charge transfer (ICT) photophysical properties. The earned compound exhibited significant polarity sensitivity, which is feasible to distinguish microenvironment polarity of the hydrophobic region in Aβ-species from those of aqueous media. Besides, we chose this particular diketone moiety because it has been previously demonstrated that this motif led to improved aqueous solubility of the half-curcumin structures consisting of dioxaborine by suppressing intermolecular π-π stacking interactions due to the inherent steric bulkiness of isobutyl moiety[25]. To this end, the quinoline-derived half-curcumin-dioxaborine (Q-OB) was prepared to detect the oligomeric $A\beta_{1-42}$ over monomer and fibrils in the complex Aβ self-assembly cascade for an early stage of AD. By suitable physicochemical properties and biocompatibility, Q-OB achieved excellent blood-brain barrier (BBB) penetrability and in vivo imaging of Aβ in AD transgenic mice. In addition, as an unprecedented attempt in small molecule sensors, this study aimed to develop a diagnosis platform using the cerebrospinal fluid (CSF) samples of AD patients to evaluate accuracy and potential for a paradigm shift of early-AD diagnosis (Fig. 1).

## Results and discussion
### Rational design, synthesis, and photophysical characterization
To achieve an excellent fluorescence response towards Aβ oligomers, Q-OB probe was rationally designed to consist of four components based on half-curcumin structure: (1) Harnessing heterocyclic *N,N*-dimethyl-2-quinolinamine providing suitable hydrophobicity for significant binding affinity to exclusively Aβ oligomers; (2) extended π-conjugation with a double-bond π-bridge serving bathochromic shift and adjusting lipophilicity for BBB penetrability; (3) a dioxaborine moiety as an electron-acceptor for intensive push-pull effect; and (4) isobutyl group providing enhanced aqueous solubility due to inhibiting intermolecular π-π stacking by steric bulkiness (see bottom panel of Fig. 1). The intermediate was synthesized via consecutive synthetic route involving lithium-halogen exchange, facilitating the transformation of 6-bromo-*N,N*-dimethyl-2-quinolinamine (1) into the desired aldehyde compound, 2-(dimethylamino)-6-quinolinecarboxaldehyde (2) as shown in Fig. 2a. Subsequently, the dioxaborine motif was

formed from 6-methylheptane-2,4-dione and following in situ aldol condensation was employed to obtain Q-OB probe with a 33.6% overall yield (Fig. 2b).

Similar to most currently used fluorophores for Aβ oligomers, Q-OB is a D-π-A-type dye exhibiting large Stokes' shifts (approximately 110 nm) with a broad absorption/emission spectrum, highly beneficial for optical imaging. Photophysical properties of Q-OB were examined, particularly its ICT behaviors with various organic solvents (Supplementary Table 2). Maximum absorption and emission wavelengths of Q-OB were obtained concerning solvent polarity index, polarizability and dipolarity as reported in the Catalán data set[37]. The electronic stabilization of the Frank–Condon state caused by solvent polarizability was well correlated with the maximum absorption wavelength. By contrast, the wavelength of maximum fluorescence emission was highly correlated to solvent dipolarity, suggesting solvent cage rearrangement, which results in geometrical relaxation at the excited states (Supplementary Fig. 10). Taken as a whole, these solvatochromic effects verify the excited state of Q-OB with a significantly stronger dipole moment than that of the ground state, consistent with the ICT characteristics of Q-OB (Supplementary Fig. 11). The fluorescence quantum yield is the highest in the most less-polar conditions (~90%) and decreases with the high polarity of solvents. The lowest quantum yield (8.2%) was observed in methanol for Q-OB, a common observation for ICT-based fluorophores[38]. These results were evaluated with density functional theory (DFT) calculation in order to support the occurrence of ICT properties of Q-OB. The distributions of electron density revealed redistribution from *N,N*-dimethyl amino quinoline moiety in the highest occupied molecular orbital (HOMO) to the dioxaborine moiety in the lowest unoccupied molecular orbital (LUMO) as shown in Supplementary Fig. 12.

To appropriately perform the fluorescence marking function as an Aβ oligomers-sensing dye and for potential applications, including in vivo imaging and clinical diagnosis using biofluids in vitro, Q-OB is required to maintain reasonable physicochemical properties, such as solubility and photostability in aqueous media. To overcome aggregation-caused quenching, an isobutyl moiety was employed in the half-curcumin core, which interfered with crystal packing to avoid intermolecular aggregation. Using concentration-dependent absorbance spectra, the enhanced water solubility that resulted from employing isobutyl moiety was confirmed. At concentrations up to 50 μM of Q-OB, an exponential relationship between concentration and absorbance was obtained, and critical micelle concentration was determined to be 28.9 μM (Supplementary Fig. 13). Additionally, Q-OB demonstrated reasonable photostability under intensive light irradiation for 12 h, which could endure long-term light irradiation in the kinetic assay of Aβ self-assembly (Supplementary Figs. 14, 15). Next, the absorbance and fluorescence spectra were determined in the presence of $A\beta_{1-42}$-species, including $A\beta_{1-42}$ monomer, fibrils, and oligomers (Fig. 2c). As shown in Fig. 2d, the maximum absorbance of Q-OB was observed at 454, 460, and 465 nm and the maximum fluorescence emission bands were determined as 567, 587, and 577 nm for $A\beta_{1-42}$, monomer, fibrils, and oligomers, respectively, with significant Stokes' shift (approximately 110 nm). We note that fluorescence response could occur through two potential mechanisms—conformational restriction and/or microenvironmental polarity changes via interaction between protein and ligand. Since fluorescence intensity barely varies as a function of increasing viscosity in binary glycerol/ethylene glycol solutions, we assumed that the microenvironmental polarity changes upon surface interaction are crucial for fluorescence turn-on with the amphiphilic (Aβ monomer and oligomers) or hydrophobic (cross-β-sheet of Aβ fibrils) sites (Supplementary Fig. 16 and Supplementary Table 3). The polarity of the environment sensed by Q-OB was found to be best matched with that of tetrahydrofuran[39]. Therefore, considering the poor quantum yield of fluorescence in an aqueous condition, Q-OB exhibited a considerably high fluorescent turn-on

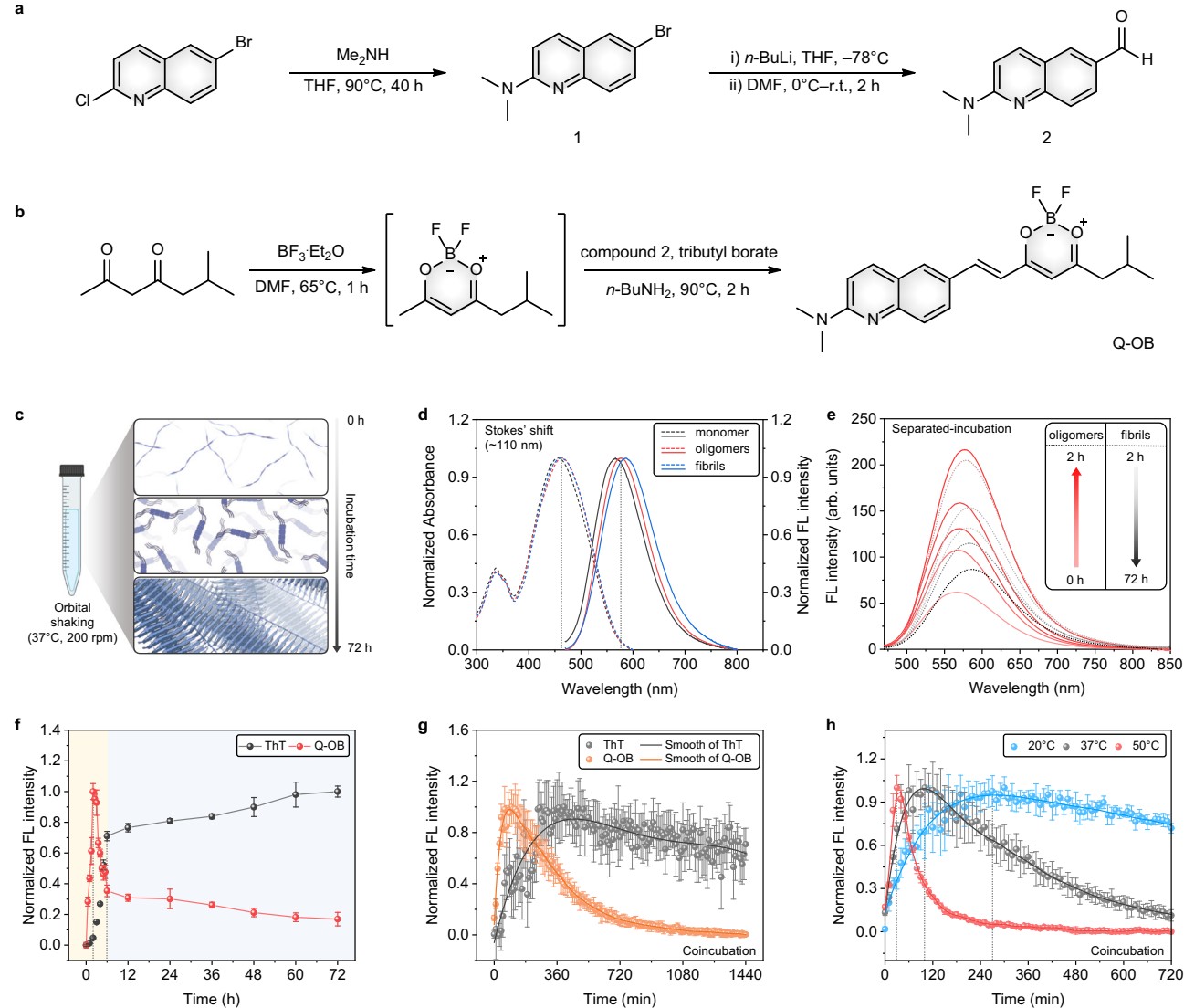

**Fig. 2 | Absorbance and fluorescence spectroscopy data of Q-OB.** Synthetic route of (**a**) compound 2 and (**b**) Q-OB. **c** Representative illustration of Aβ-species formation. **d** Normalized absorption and fluorescence emission spectra of Q-OB (5 µM) in the presence of Aβ₁₋₄₂-species, including monomer, oligomers, and fibrils (10 µM) in phosphate-buffered saline (PBS) (10 mM, pH = 7.4). **e** Fluorescence spectra for identifying Aβ aggregation using Q-OB under the separated-incubation. **f** Time-dependent plot for the change in the fluorescence intensity of Q-OB and thioflavin T (ThT) in the presence of Aβ₁₋₄₂. Data are presented as mean values ± standard deviation (s.d.) derived from $n = 3$ independent experiments. **g** Identification of Aβ aggregation kinetics using Q-OB and ThT under the coincubation at 37 °C, and (**h**) other temperatures (20 and 50 °C) with orbital-shaking (200 rpm). Data are represented as mean ± s.d. with $n = 3$ independent experiments. The abbreviated word is FL, fluorescence. Panel **a** was created with BioRender.com. Source data underlying **d**–**h** are provided as a Source Data file.

ratio through interaction with hydrophobic binding sites of oligomeric Aβ₁₋₄₂.

Time-dependent fluorescence response with Aβ self-assembly kinetics was obtained employing Q-OB and universal reference dyes for detecting Aβ fibrillogenesis in vitro; thioflavin T (ThT) and 8-anilino-1-naphthalenesulfonic acid (ANS). Two different experimental conditions (i.e., separated-incubation and coincubation) in phosphate-buffered saline (PBS) (20 mM, pH = 7.4) were applied to investigate the fluorescence response of Q-OB towards Aβ-species in kinetic assay. Under the separated-incubation, the fluorescence intensity of Q-OB rapidly increased during the incipient state of Aβ self-assembly up to 2 h (red-solid line), followed by a decrease in the intensity during the matured aggregates formed for up to 72 h (gray-dotted line) as shown in Fig. 2e and Supplementary Fig. 17. In contrast, the fluorescence intensity of ThT increased for 6 h, followed by marginally rising for 72 h with matured Aβ aggregates formation, since ThT specifically detects cross-β-sheet via hydrophobic binding (Fig. 2f and Supplementary

Fig. 18). Meanwhile, ANS demonstrated non-specific responses to certain Aβ-species in contrast to Q-OB or ThT because the sensing mechanism of ANS based on electrostatic interactions to cationic groups of Aβ-species is not feasible to discern certain Aβ-species during amyloid fibrillogenesis (Supplementary Fig. 19).

Subsequently, coincubation kinetics of Aβ self-assembly was measured using ThT, ANS, and Q-OB to evaluate whether coincubation in a mixed state affects the aggregation mechanism of Aβ. For Q-OB, the fluorescence intensity drastically increased as oligomeric Aβ₁₋₄₂ was concentrated and then decreased with matured aggregate formation (Fig. 2g and Supplementary Fig. 20). Whereas the fluorescence intensity of ThT increased with a maturity of fibrillar Aβ₁₋₄₂ aggregates for approximately 360 min, followed by a slight decrease in the intensity due to photobleaching. The fluorescence response of ANS tended to decrease continuously with the non-specific response towards certain Aβ-species (Supplementary Fig. 20a, b). Furthermore, artificial cerebrospinal fluids (aCSF) media, mimicking the CSF

microenvironment, was utilized to pre-examine an experiment for applying Q-OB in patients' CSF. Interestingly, aCSF condition led to the different fluorescence signals of ThT and Q-OB towards various Aβ-species in contrast to those of PBS (Supplementary Fig. 20c). As a negative control, deionized water (DW) condition demonstrated the nonapparent Aβ fibrillogenesis in Q-OB and ThT fluorescence response (Supplementary Fig. 20d). To further assess the kinetics results, the temperature was varied to see any different kinetics of the secondary nucleation mechanism of Aβ aggregation. Since it is known that the nucleation rate typically depends on temperature[40,41], we expected that the different Aβ aggregation kinetics could be finely traced with Q-OB during time-dependent fluorescence intensity changes. Interestingly, increased temperature (50 °C) accelerated the rate of Aβ nucleation, which showed a peak time point of 30 min, whereas low temperature (20 °C) retarded the assembly kinetics, which was observed as a delayed fluorescence change (peak time point of 270 min) as can be seen from Fig. 2h (full-time range data are described as a Supplementary Fig. 21). These tendencies, however, were not observed by Aβ$_{1-40}$; indeed, the Q-OB response demonstrated a tendency similar to that of ThT (Supplementary Fig. 22). Collectively, these kinetic differences confirm the distinguished response of Q-OB to Aβ oligomer formation in the early stages of the Aβ aggregation process.

To elucidate conformation of Aβ-species that Q-OB may recognize, we carried out biophysical characterization during its fibrillogenesis. For each Aβ$_{1-42}$-species, transmission electron microscopy (TEM) images were taken during 24 h of incubation. TEM images demonstrated that prefibrillar oligomeric structures were formed as short fibers after 2 h of incubation in PBS. Besides, it is confirmed that aggregation of prefibrillar oligomers led to protofibril formation approximately 6–12 h after, and matured fibrils were produced within

24 h (Fig. 3a). Further, the morphology of Aβ$_{1-42}$-species in aCSF was investigated, and spherical aggregates were observed after 2 h of incubation (Supplementary Fig. 23). Both incubation conditions induced an entangled form of oligomer agglomeration after approximately 6 h of incubation. Subsequently, the oligomer agglomeration typically matured into fibrils in PBS condition, while we observed that gradually enlarged oligomer agglomeration occurred only in the aCSF condition. Simultaneously, circular dichroism spectroscopy was carried out to examine the secondary structure. According to the result, monomeric Aβ$_{1-42}$ exhibited a random coil when freshly initiated, consistent with previous research findings[42]. It is interesting to note that both Aβ$_{1-42}$ oligomers and monomer were observed to possess random coil content, while Aβ$_{1-42}$ fibrils had a typical β-sheet structure (Supplementary Fig. 24). These biophysical characterization results indicate not only accurately sampled Aβ but also indirectly suggest why Q-OB exhibited significantly different photophysical properties in various Aβ-species in aCSF (or CSF) compared to those in PBS.

Relatively predominant recognition of Q-OB toward Aβ oligomers was confirmed using comprehensive Aβ-species and potential analytes. Note that Aβ$_{1-42}$ oligomers enhanced the fluorescence intensity by 102.87- (blank), 26.61- (Aβ$_{1-40}$ monomer), 4.38- (Aβ$_{1-42}$ monomer), 4.62- (Aβ$_{1-40}$ fibrils), and 2.51-folds (Aβ$_{1-42}$ fibrils) in PBS, which suggested that the interaction of Q-OB with Aβ$_{1-42}$ oligomers is comparatively effectual in achieving significant fluorescence emission over others (Fig. 3b and Supplementary Fig. 25). In addition, the fluorescence response of Q-OB to potential interferants, such as metal ions, amino acids, and thiols, was negligible compared to that of Aβ oligomers (Supplementary Fig. 26). Moreover, Q-OB demonstrated a relatively slight fluorescence increase with protein interferants such as human serum albumin (HSA), bovine serum albumin (BSA), and phosphorylated tau (p-Tau), typical interferants for a hydrophobic

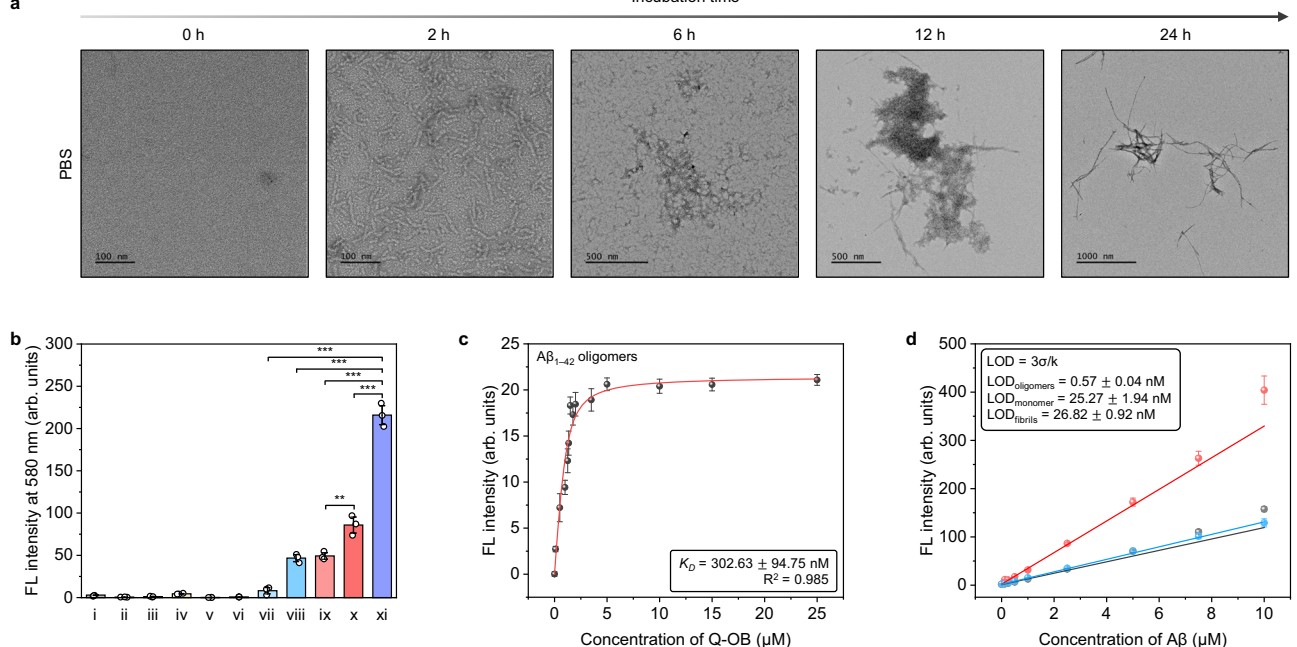

**Fig. 3 | Photophysical properties of Q-OB as an Aβ oligomers-sensing probe.** **a** Biophysical characterization using transmission electron microscopy (TEM) of Aβ-species at specific time points (0, 2, 6, 12, and 24 h) of fibrillogenesis incubated in PBS (20 mM, pH = 7.4). Scale bars indicate the length stated in the figure. **b** Fluorescence intensity of various Aβ-species (10 μM, i: blank; ii: Aβ$_{1-40}$ monomer; iii: Aβ$_{1-40}$ fibrils; iv: Aβ$_{1-42}$ monomer; v- Aβ$_{1-42}$ fibrils; vi: Aβ$_{1-42}$ oligomers) and Q-OB (5 μM) with Aβ-species (10 μM, vii: Aβ$_{1-40}$ monomer; viii: Aβ$_{1-40}$ fibrils; ix: Aβ$_{1-42}$ monomer; x: Aβ$_{1-42}$ fibrils; xi_ Aβ$_{1-42}$ oligomers). Data points with error bar stands for mean s.d. derived from $n$ = 3 independent experiments. The $p$-values were

generated with two-sided unpaired $t$-test; *$p$ < 0.05, **$p$ < 0.01, ***$p$ < 0.001. The exact $p$-values are provided in a Source Data file. **c** Saturation binding curves of Aβ$_{1-42}$ oligomers (10 μM) as a function of Q-OB (0–25 μM). Error ranges represent s.d. ($n$ = 3, independent experiments). **d** A linear correlation between fluorescence intensity of Q-OB (1 μM) at various concentrations (0–10 μM) of Aβ$_{1-42}$ monomer, oligomers, and fibrils. Error bars represent s.d. from $n$ = 3 independent experiments (slit 2.5/2.5). The abbreviated words are FL, fluorescence; LOD, limit of detection. Source data are underlying (**b–d**) provided as a Source Data file.

fluorescent dye. Absorbance and fluorescence emission spectra were obtained in the pH range of 1–11; results rarely showed a change in the pH range of 4–11, but sharp changes were detected at pH under 3 (Supplementary Fig. 27). This observation indicates that Q-OB is pH-independent in fluorescence emission under even a physiological environment. Furthermore, fluorescence titration of Q-OB revealed its binding affinity for oligomeric $A\beta_{1-42}$, with the dissociation constant ($K_D$) of 302.63 ± 94.75 nM, while insignificant binding affinities were noted with $A\beta_{1-42}$ monomer and matured fibrils ($K_D$ = 4745.18 ± 722.56 and 5667.33 ± 1801.53 nM, respectively) (Fig. 3c and Supplementary Fig. 28). To examine the sensitivity of Q-OB to $A\beta_{1-42}$ oligomers, the limit of detection was calculated by measuring fluorescence intensity of Q-OB with various concentrations of each $A\beta_{1-42}$-species, and the detection limits were found to be 0.57 ± 0.04, 25.27 ± 1.94, and 26.82 ± 0.92 nM for oligomers, monomer, and fibrillar aggregates, respectively, as shown inset to Fig. 3d.

In addition, the specificity of the Q-OB was estimated to $A\beta_{1-42}$ oligomers and potential interferents by assessing cross-reactivity in an aqueous media. The fluorescence intensity of Q-OB was recorded with $A\beta_{1-42}$ oligomers in the presence of cross-reactive interferents, including $A\beta_{1-40}$ or $A\beta_{1-42}$-species (i.e., monomer and fibrils) prepared in various concentration ratios of $A\beta_{1-42}$ oligomers (×0 (blank), ×1, ×10, ×100, and ×500) in PBS. In each experimental condition, the fluorescence intensity of Q-OB exclusively changes with only $A\beta_{1-42}$ oligomers while not changed in the presence of cross-reactive interferents. As a result, Q-OB is negligibly cross-reactive towards even more than 100-times concentrated both $A\beta_{1-40}$ monomer and fibrils (Supplementary Fig. 29a, b). In contrast, Q-OB was found to be cross-reactive to both $A\beta_{1-42}$ monomer and fibrils as seen in Supplementary Fig. 29c, d. Furthermore, to evaluate the influence on oligomerization kinetics of $A\beta_{1-42}$ in the presence of interferents, a coincubation kinetic assay was implemented using Q-OB, with certain amyloid proteins ($A\beta_{1-40}$ and p-Tau) and albumins (HSA and BSA). Interestingly, those interferants negligibly influenced sensing formation of prefibrillar $A\beta_{1-42}$ oligomers with Q-OB. The marginal increment of fluorescence intensity of Q-OB towards analyte mixture of $A\beta_{1-40}$ and $A\beta_{1-42}$ might be attributed to signal amplification by promoting oligomerization of $A\beta_{1-40}$ by $A\beta_{1-42}$ (Supplementary Fig. 30a)[5]. p-Tau exhibited a negligible change of fluorescence response, as shown in Supplementary Fig. 30b, and albumins induced an earlier peak of fluorescence intensity for the oligomer-induced signal of Q-OB, approximately 20 min (Supplementary Fig. 30c, d). Collectively, the fluorescence change of Q-OB is entirely independent of those potential interferants, confirming again that Q-OB is exclusively responsive to $A\beta_{1-42}$ oligomers over other competitive amyloid proteins as a potential application using biofluids in vitro.

## Identification of molecular interactions

To better understand the interaction between $A\beta$-species and Q-OB, we conducted an $A\beta$ fragment mapping assay. The mapping assay that determines specific residues interacting with $A\beta$-probes was previously reported using a mapping amyloid plate[43]. 37 hexamer $A\beta_{1-42}$ fragments, from 1–6 to 37–42, were immobilized on a 96-well plate and incubated with Q-OB for 24 h. Then the fluorescence values in each well of the mapping amyloid plate at the emission wavelength of Q-OB for baseline values after washing out to remove unbound dyes (Supplementary Fig. 31). In consequence, four $A\beta$ fragments; $A\beta_{14-19}$, $A\beta_{15-20}$, $A\beta_{16-21}$, and $A\beta_{37-42}$, apparently elevated the fluorescence response of Q-OB compared to that of a full-length $A\beta_{1-42}$ (Fig. 4a). The responsive fragments comprise the major sequences in the hydrophobic central ($A\beta_{13-20}$, HHQKLVFF) and C-terminal ($A\beta_{31-42}$, IIGLMVGGVVIA) regions. The central region KLVFF ($A\beta_{16-20}$) has been previously reported as the interacting domain of $A\beta$ with curcumin analogs[44–46]. To visualize the spectrum of the binding sites of Q-OB on a full-length $A\beta_{1-42}$ sequence, a heatmap with a two-color gradient

(white to red, 0–100%) is shown in Fig. 4b. The spectral value of each amino acid was obtained by averaging fluorescence intensities of $A\beta$ fragments containing the corresponding amino acids. In addition, the fluorescence intensity of Q-OB with an additional coincubation process with $A\beta_{1-42}$ was measured. The addition of $A\beta_{1-42}$ to mapping amyloid plate induced non-specific changes in the fluorescence pattern of Q-OB (Supplementary Fig. 32).

Collectively, these results indicate that the immobilized $A\beta_{1-42}$ peptide fragments remained in monomeric, aggregation-prone states and were readily recognized by Q-OB, which can identify potential binding sites of Q-OB molecules toward oligomeric $A\beta_{1-42}$. Indeed, Q-OB mostly interacted with the central hydrophobic (14–21) and C-terminal region (29–42), which play important roles in aggregation[47]. The central hydrophobic region comprising KLVFFA is known to be the most aggregation-prone sequence in $A\beta_{1-42}$, enable to generate fibrils itself[48]. Hydrophobic C-terminal region also tends to participate in $A\beta$ fibrils formation and stabilizes $A\beta$ oligomers[49,50]. Additionally, the cryo-EM study, which determined the structure of $A\beta$, revealed that these two regions are the key segments of fibrillar $A\beta$[51]. Therefore, we hypothesize that the predilection for $A\beta$ oligomer exhibited by Q-OB is attributed to the transient occurrence of exposed hydrophobic regions on the surface of oligomeric $A\beta$ intermediates, preceding the assembly into mature fibrils characterized by obstructed binding sites.

Furthermore, we utilized a molecular weight cut-off filter to narrow down the target $A\beta_{1-42}$ oligomers range of Q-OB. Using a 100 kilodaltons (kDa) cut-off filter, we separated low- and high-molecular-weight oligomers to investigate which oligomer size comprises a major part of the dynamic mixture. In low-molecular-weight oligomers (<100 kDa), fluorescence intensity of Q-OB gradually decreased from the initial point, which might have resulted in the highest concentration of oligomers before the mature fibril state (Fig. 4c). In high-molecular-weight oligomers (>100 kDa), a 2.1-fold higher fluorescence intensity was observed than that in the low-molecular-weight oligomers at the initial point, following decreased similar to the tendency of low-molecular-weight (Fig. 4d). Predictably, ThT showed gradually increasing fluorescence intensity in the same experimental process, which indicated that the decayed fluorescence emission of Q-OB with both-size of $A\beta$ oligomers was induced by the maturity of $A\beta$ fibrils (Fig. 4e). These results demonstrated that Q-OB predominantly interacted with $A\beta_{1-42}$ oligomers larger than 100 kDa, indicative of assemblies exceeding an icosikaidimer, 22-monomer assembly.

## Brain tissue imaging of 5xFAD transgenic mice and AD patients

Motivated by the in vitro findings, our investigation delved into detection capabilities of Q-OB for $A\beta$ aggregates in biological samples through a set of brain tissue imaging experiments. For immunofluorescence analysis, we employed whole brain tissues obtained from 12- and 16-month-old 5xFAD transgenic AD model mice, alongside 8-month-old B6 wild-type mice as controls. The co-staining of each brain tissue with Q-OB and an anti-$A\beta$ antibody, 6E10, revealed pronounced $A\beta$ accumulation. Concentration-dependent brain tissue imaging demonstrated the fluorescent labeling of 5xFAD transgenic AD mice using Q-OB dye, with optimal staining achieved at a modest concentration of 1 μM. Notably, there was no observed unspecific staining of the white matter (Supplementary Fig. 33). Further examination using Q-OB staining revealed dot-shaped extracellular amyloid deposits in 5xFAD mice brain tissues at green/red channels. However, significant autofluorescence in red channel, attributed to lipofuscin, was observed across the entire tissue section (Supplementary Fig. 34, 35)[52]. Despite this interference, Q-OB staining pattern exhibited a conspicuous co-localization with 6E10-positive stains in the comprehensive plaque area (upper panels of Fig. 5a and Supplementary Fig. 36). Q-OB-labeled $A\beta$ deposits exhibited co-localization with 6E10-stained $A\beta$ accumulations, while such co-localization in the red channel was interfered by autofluorescent factors (Supplementary Fig. 37).

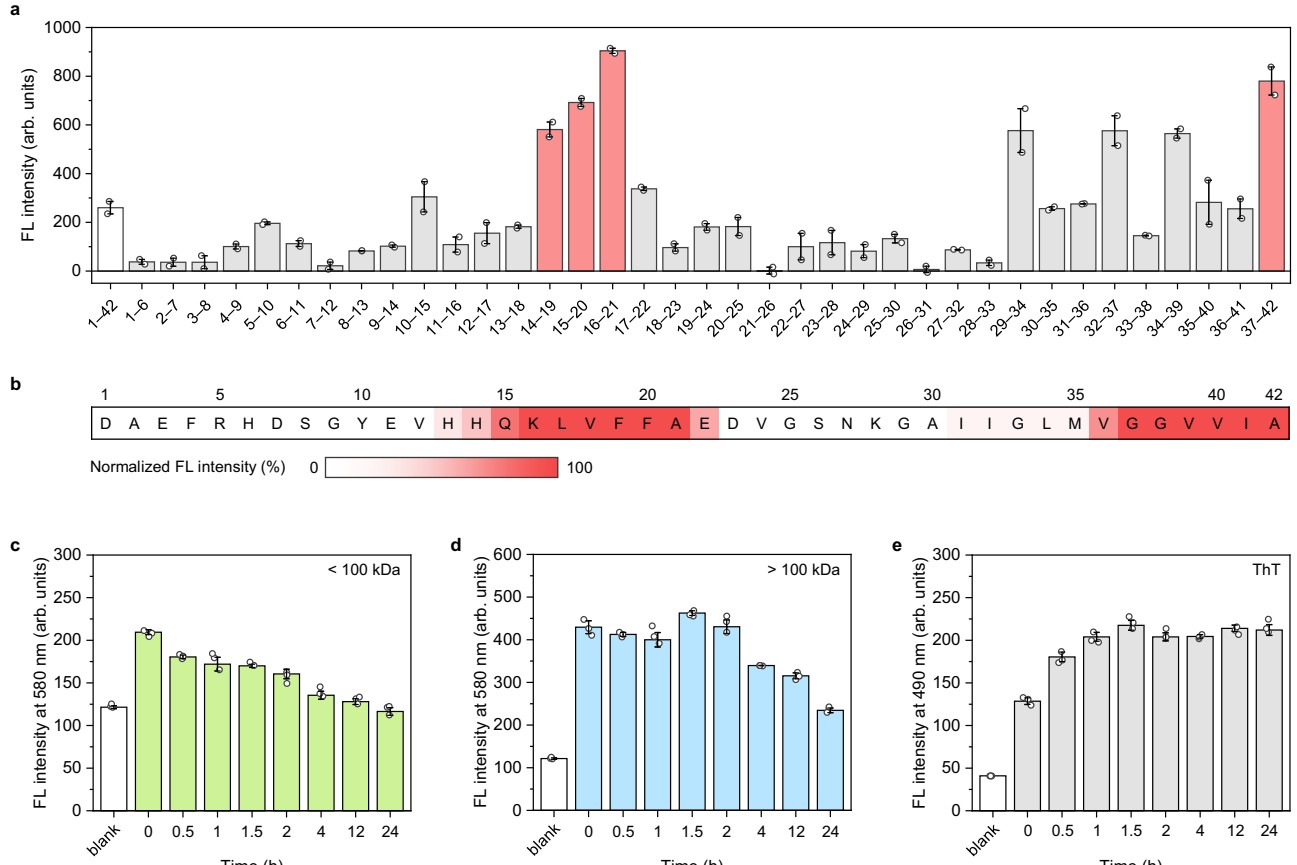

**Fig. 4 | Identification of binding residues of Q-OB. a** Identification of Q-OB binding sites for Aβ$_{n-(n+5)}$ fragments of a full-length Aβ$_{1-42}$. Data of fluorescence values of Q-OB, and (**b**) the heatmap displays the spectrum of binding intensity on Aβ$_{n-(n+5)}$ sequence (white to red, 0–100%) ($\lambda_{ex}/\lambda_{em}$ = 500/620 nm). Data points with its error range stands for mean s.d. derived from $n$ = 2 independent experiments. Time-dependent fluorescence intensity of Q-OB with Aβ$_{1-42}$ pre-treated molecular weight cut-off of filter, including (**c**) under 100 kilodaltons (kDa), (**d**) over 100 kDa ($\lambda_{ex}/\lambda_{em}$ = 490/620 nm) (mean ± s.d., $n$ = 3 independent experiments). **e** Time-dependent fluorescence intensity of ThT with Aβ$_{1-42}$ ($\lambda_{ex}/\lambda_{em}$ = 450/480 nm) (mean ± s.d., $n$ = 3 independent experiments). The abbreviated words are FL, fluorescence; ThT, thioflavin T. Source data are provided as a Source Data file.

Given that these mice were subjected to perfusion to remove blood before the brain extraction, it is suspected that soluble Aβ oligomers were also withdrawn from the brain and, thus, not only Q-OB but also the pan-Aβ 6E10 antibody would lose a chance to stain oligomers. Hence, these results indicated that Q-OB dye is not restricted to the detection of Aβ oligomers in transgenic AD mice brains.

Next, postmortem cerebral hippocampal tissue specimens were obtained from a total of 13 individuals for histochemical analysis to determine whether Q-OB could stain Aβ aggregates in the human brain tissues (Supplementary Table 4). Formalin-fixed and paraffin-embedded tissues were deparaffinized with xylene, and each brain tissue was stained by Q-OB and anti-Aβ antibody (6E10) under the same conditions as the mice brain tissue imaging. Q-OB could mark senile plaques in the hippocampal area of brain tissue from AD-diagnosed patients in green channel, similar to the mice tissue (bottom panels of Fig. 5a and Supplementary Fig. 38). Negligible unspecific binding in the white matter was found on the brain tissue of cognitive normal (CN) groups, as a nondemented control, while an inadvertent autofluorescent factor was observed in green/red channels, thereby impeding the unambiguous identification of amyloid deposit-stained by Q-OB dye (Supplementary Fig. 39). Furthermore, in contrast to the imaging of mice brain tissues, Q-OB-stained Aβ aggregates predominantly manifested in the green channel, revealing distinct staining patterns in comparison to the red channel images. Q-OB-stained senile plaque regions exhibited extensive co-localization with 6E10-stained Aβ deposits (most of the senile plaques) in the green channel, while such co-localization near senile plaques in the red channel was infrequent (Supplementary Fig. 40). As shown in Fig. 5b and Supplementary Fig. 41, pseudo-color merge of stained Aβ aggregates using Q-OB and anti-Aβ antibody marked by hollowed arrow (overlap of green and blue channels), and yellow arrows indicate autofluorescent factor. Moreover, Q-OB labeled cerebral amyloid angiopathy by demonstrating the accumulation of Aβ within cerebral blood vessels in green channel merged with 6E10 (Supplementary Fig. 42). It is evident that the somewhat clear labeling of cerebral amyloid angiopathy by Q-OB is not limited to species of Aβ, especially oligomers, and that fibrillar deposits can also be stained inclusively.

To explore the feasibility of Q-OB for in vivo optical imaging applications, cytotoxicity/phototoxicity of Q-OB was evaluated on a human-derived neuroblastoma cell line (SH-SY5Y) in vitro. For this, SH-SY5Y cell was preincubated with Q-OB, and cell viability was examined with or without light irradiation. Q-OB showed negligible toxicity to SH-SY5Y under both conditions at 0–100 μM concentration ranges for 24 h incubation (Supplementary Figs. 44, 45). Parallel artificial membrane permeability assay (PAMPA) and computational calculation verified the excellent BBB penetrability of Q-OB (Supplementary Fig. 46 and Supplementary Table 6). Indeed, Q-OB was effectively localized into the brain after intravenous injection of Q-OB into B6 wild-type mice ex vivo, which indicates that Q-OB could be utilized as a potential imaging agent of the brain biomarkers (Supplementary

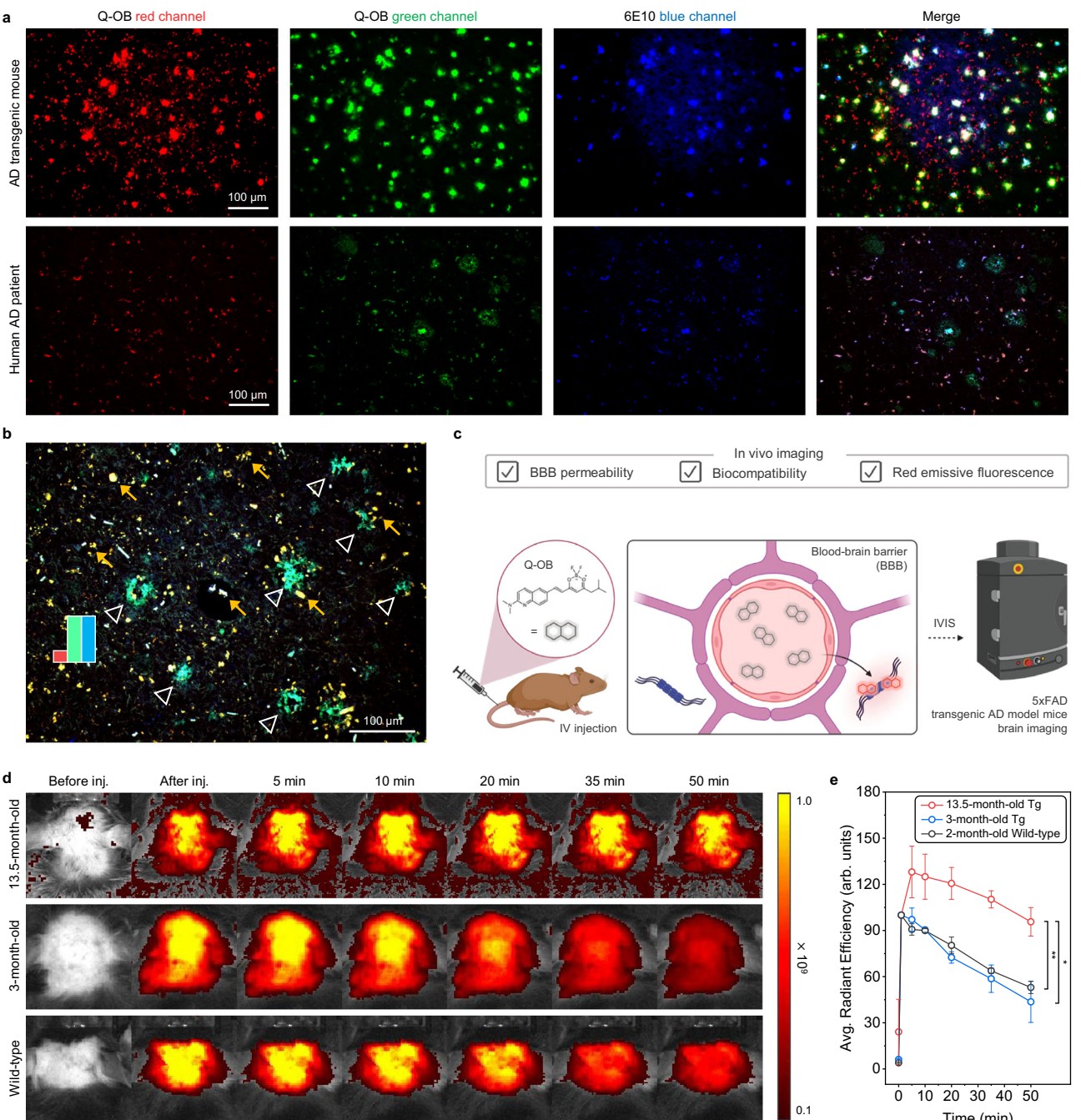

**Fig. 5 | In vitro and In vivo imaging of Alzheimer's disease (AD) brain tissue sections. a** Fluorescence images of cortex area in the brain tissue slice from 16-month-old 5xFAD transgenic AD model mouse (upper panels) and hippocampal area of AD patients' brain tissue (bottom panels) stained by Q-OB (1 μM, red and green channels) and 6E10 (anti-Aβ antibody, 1:200, blue channel) (red channel; $\lambda_{em}/\lambda_{ex} = 520/590$ nm, green channel; $\lambda_{em}/\lambda_{ex} = 480/520$ nm, and blue channel; $\lambda_{em}/\lambda_{ex} = 360/470$ nm); Scale bar: 100 μm. **b** Pseudo-color overlapping of stained Aβ aggregates in human AD patient brain tissue marked by white-hollowed arrows (overlap of green and blue channels) and yellow arrows indicate autofluorescence; Scale bar: 100 μm. **c** Representative illustration of the blood-brain barrier (BBB) penetrability of Q-OB for the brain in vivo imaging. **d** Representative fluorescence signal of in vivo imaging and (**e**) average radiant efficiency, 50 min subsequent to tail vein injection of Q-OB to transgenic AD model mouse brains, including 13.5- and 3-month-old 5xFAD, and 2-month-old B6 wild-type mice. Error bars indicate s.d. (*n* = 3 biologically independent animals). The *p*-values were obtained with two-sided unpaired *t*-test; \**p* < 0.05, \*\**p* < 0.01. The exact *p*-values are provided in a Source Data file. The fluorescence images were captured with an IVIS imaging system at an excitation wavelength of 500 nm, and the emission was monitored at 620 nm. The abbreviated words are IV, intravenous; Tg, transgenic. Panel **c** was adapted from "Blood Brain Barrier (Transverse)," by BioRender.com (2023). Retrieved from https://app.biorender.com/biorender-templates. Source data underlying (**e**) are provided as a Source Data file.

Fig. 47). Based on excellent BBB permeability, biocompatibility, and red fluorescence-emissive photophysical properties of Q-OB, a non-invasive imaging setup; the in vivo imaging system (IVIS) was used to perform image acquisition for visualizing Aβ deposits in vivo (Fig. 5c). 3- and 13.5-month-old 5xFAD transgenic AD model and 2-month-old B6

wild-type mice were prepared, and Q-OB solution was injected intravenously. In 3-month-old AD mice, the fluorescence signal of the brain region increased after injecting Q-OB through the tail vein, and the intensity decreased afterward for 50 min (Fig. 5d and Supplementary Fig. 48). The fluorescence pattern in 3-month-old AD mice was similar

to that in the wild-type mouse after injection of Q-OB. In 13.5-month-old AD mice, the fluorescence stronger intensity of Q-OB was observed than those of 3-month-old AD mice (1.3-folds) and wild-type (1.4-folds) at 5 min, and these gaps remained significant even 50 min after the injection (2.2- and 1.8-folds for 3-month-old AD and wild-type, respectively) (Fig. 5e).

Given that Aβ production starts at 1.5 months and its extracellular deposition can be observed after 2 months in the 5xFAD transgenic mouse[52], this observation is reasonable to demonstrate that the disposed of Aβ in 3-month-old mice was insufficient to be detected while older mice possessed sufficient Aβ that could interact with Q-OB. Nevertheless, the results of this optical bioimaging should be interpreted with some level of caution, as it is still constrained for in vivo assay. It is still challenging to differentiate between prodromal (young) AD and older transgenic AD model mice using optical imaging to target Aβ oligomers due to limiting a specific Aβ state in such a physiological environment, especially for oligomers. Besides, the probe for further optical bioimaging should meet several additional requirements, including pharmacokinetics (logical washout kinetics) and in vivo toxicity (low physiological toxicity). Therefore, these results briefly suggested the possibility of in vivo application of Q-OB to measure the number of Aβ-species in the brain regions using transgenic AD model live mice.

## Clinical application for in vitro early-AD diagnosis using CSF of patients

Biomarkers found in the CSF of AD patients are being explored as diagnostic criteria for research and clinical trials to improve the accuracy of antemortem diagnosis in vitro[4,53]. Major CSF biomarkers of AD, including Aβ$_{1-42}$, p-Tau$_{181}$, and total tau (t-Tau), can be used to identify pathogenic changes in the brain during AD (i.e., axonal and synaptic degeneration, Aβ plaque deposition, and neurofibrillary tangles)[54–56]. Hence, assays for detecting these proteins using CSF samples from AD patients and healthy controls can be employed for diagnostics assessments[57]. As CSF biomarkers can be detected during the early stage of AD, even before symptoms of incipient dementia emerge, it is possible to discriminate patients with prodromal AD, who often progress to dementia, compared to healthy controls[58–62]. A collection of AD-related neuropathological alterations is provided in the most recent NIA-AA research framework[62]. This definition establishes criteria based on postmortem examination of the affected brain and by using in vivo A/T/N (ATN) biomarkers rather than clinical symptoms. In this classification system, the value of an A (i.e., Aβ$_{1-42}$ or amyloid PET) is referred to as an Aβ biomarker. T denotes the significance of a biomarker for tau pathology, p-Tau$_{181}$. N represents neurofilament light chain (NFL) as neuronal damage or neurodegenerative biomarkers, including t-Tau[63].

Therefore, according to these diagnostic indexes, clinical applications using Q-OB aimed to establish a diagnostic system (henceforth referred to as Q-OB assay), compared to a commercially available analytical platform, Lumipulse fully automated immunoassay and two manual immunoassays (INNOBIA AlzBio3 xMAP), with respect to accuracy and the potential for early diagnosis of AD (Fig. 6a)[64]. For this analysis, CSF samples of 61 subjects were used, and individuals were classified based on multiple diagnostic analytes, including the concentrations of ATN CSF biomarkers obtained by INNOBIA AlzBio3 xMAP and amyloid PET imaging evaluation. Besides, z-scores were estimated to assess the individual cognitive performance on the neuropsychological domain using the Seoul Neuropsychological Screening Battery (SNSB)—a comprehensive neuropsychological test that assesses five cognitive domains: attention, language, visuospatial, memory, and frontal[65]. As a result, cognitive and neuropsychological scores differed significantly among the groups, with descending performance among CN ($n = 22$) as a nondemented control group, mild cognitive impairment (MCI, $n = 21$), and Alzheimer's disease dementia

(ADD, $n = 18$), and the CSF biomarker levels among immunoassay were strongly intercorrelated. The demographic data of absolute values for Aβ$_{1-42}$, p-Tau$_{181}$, and t-Tau showed that the concentration of Aβ$_{1-42}$ decreased while that of p-Tau and t-Tau increased, according to the AD pathological cascade. In particular, a significant difference was noted in the p-Tau levels of the CN and MCI groups (Table 1).

Coincubation fluorescence kinetic assay was conducted for CN, MCI, and ADD CSF samples using Q-OB with exogenously added full-length monomeric Aβ$_{1-42}$. Interestingly, the fluorescence intensity of Q-OB apparently increased as the aggregate was formed in the CN samples for approximately 2 h, verifying the fluorometric detection of Aβ$_{1-42}$ oligomerization in the CSF mixture (Supplementary Fig. 49). In contrast, the MCI and ADD samples showed relatively meager fluorescence intensity changes than CN under the same experimental conditions using Q-OB (Supplementary Figs. 50, 51). We carefully postulate that these distinct Aβ kinetics in each CSF sample may be caused by different concentrations of endogenous Aβ$_{1-42}$. However, this hypothesis needs to be interpreted cautiously, as we could not suggest a direct rationale for observing Aβ$_{1-42}$ oligomer formation in patients' CSF and molecular interaction of Q-OB. Furthermore, the significant heterogeneity of human CSF may cause inaccuracy and poor reproducibility in the repeated measurement process and analysis.

In order to precisely compare the variance in the fluorescence intensity, the logarithm intensity ratios between the initial (I$_0$) and 2 h-incubated (I) points were calculated, and log ($I/I_0$) values were obtained as 0.34 ± 0.13 (CN), 0.15 ± 0.12 (MCI), and 0.14 ± 0.10 (ADD), respectively, can be shown in Fig. 6b. These values show that a clinical diagnosis can be achieved using the Q-OB assay by distinguishing the CN and/or MCI/ADD states of AD patients using CSF. Based on our experimental data showing a negligible difference in log ($I/I_0$) values between MCI and ADD, it was not feasible to distinguish MCI from ADD. However, there was a marked difference between CN and other two pathological states (MCI and ADD), which suggests that our method can diagnose the pathological progression from CN to MCI in clinical applications. In addition, we have used scatter plots to evaluate the correlation between concentrations of ATN CSF biomarkers and log ($I/I_0$) values. For the CSF Aβ$_{42}$, the plot showed a proportional correlation, while CSF p-Tau$_{181}$ and NFL exhibited an inversely proportional correlation among the diagnostic groups (Fig. 6c–e). In particular, the Q-OB assay showed similar diagnostic accuracies using INNOBIA AlzBio3 xMAP for Aβ$_{1-42}$ and p-Tau levels in CN and MCI.

Collectively, these results suggest that the concentration of the intrinsic amyloid CSF biomarker affects the Aβ fibrillogenic kinetics in the Q-OB fluorescence response. Therefore, it is noteworthy that the Q-OB assay can be a useful tool for analyzing ATN biomarkers in CSF to develop accurate and useful methods for early-AD diagnosis. Further, we believe the development of an easy-to-use diagnosis kit would help overcome limitations associated with conventional AD diagnosis methods, such as cognitive assessment questionnaires, functional behavioral assessments, and PET imaging (Fig. 6f). Our next study plans involve utilizing Q-OB analogues as diagnostic platforms for clinical applications, satisfying high inter-laboratory reproducibility, short operating time, and ease of use for a layperson, to create an efficient diagnostic assay with expanded clinical implementation as a standard AD early diagnostic method.

In summary, a significant factor in AD is the lack of accurate diagnostic methods for the preclinical or prodromal stage. Hence, precise detection of Aβ oligomers and knowledge about its emergence, concentration, and evolution in self-aggregation kinetics are indispensable for a detailed understanding of neurodegenerative diseases such as AD, and for developing various diagnostics and clinical intervention strategies, especially for early-stage diseases. For efficient early diagnosis of AD, it is desirable to discover an effectual

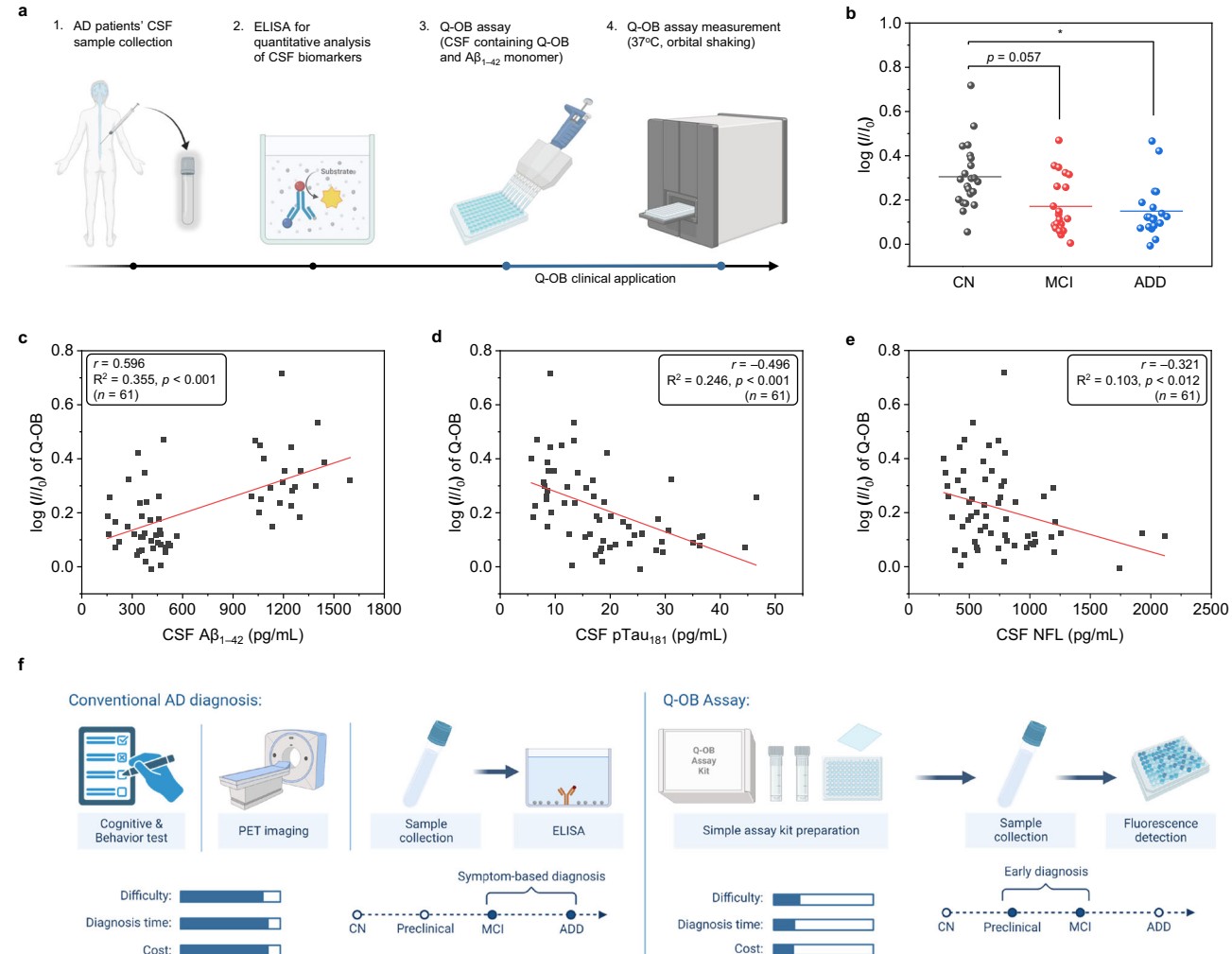

**Fig. 6 | Concentrations of cerebrospinal fluid (CSF) biomarkers in the AD cascade. a** Experimental scheme of a fluorometric kinetic assay using Q-OB for CSF samples of patients with AD and nondemented control. **b** Fluorescence intensity ratio between the initial and incubated mixtures (37 °C, 200 rpm for 2 h) of Q-OB-treated CSF samples, including cognitive normal (CN, $n = 22$ individually independent samples examined over 3 independent experiments), mild-cognitive impairment (MCI, $n = 21$ individually independent samples examined over 3 independent measurements); Alzheimer's disease dementia (ADD, $n = 18$ individually independent samples examined over 3 independent measurements). The $p$-values were generated with two-sided unpaired $t$-test; $\cdot p < 0.05$. The exact $p$-values are

described in a Source Data file. Comparison of the concentrations of CSF (**c**) A$\beta_{1-42}$, (**d**) phosphorylated tau (p-Tau$_{181}$), and (**e**) neurofilament light chain (NFL) determined using Lumipulse fully automated immunoassay and two manual immunoassays (INNOBIA AlzBio3 xMAP) and log ($I/I_0$) of Q-OB assay. The $p$-values were obtained by ANOVA with post-hoc Games-Howell test; $p < 0.001$ (**c**) and (**d**), $p < 0.012$ (**e**). The exact $p$-values are provided in a Source Data file. **f** Representative illustration of clinical applications and advantages of Q-OB assay in comparison with those of conventional AD diagnosis. Panels (**a**) and (**f**) are created with BioRender.com. Source data underlying (**b**–**e**) are provided as a Source Data file.

fluorescence probe that can selectively detect oligomeric Aβ, which is indeed medically applicable in the aspect of cost-effectiveness, rapid response, and convenience. In this context, we developed a fluorescent probe, quinoline-derived half-curcumin-dioxaborine (Q-OB), for Aβ oligomer detection in AD. Harnessing heterocyclic biannulated motif allowed for optimization of remarkable binding affinity ($K_D = 302.63 \pm 94.75$ nM), and sensitivity (LOD = $0.57 \pm 94.75$ μM) of Q-OB towards Aβ compared to Aβ monomer or fibrils during amyloid fibrillogenesis. Fragment mapping assay and molecular weight cut-off filter exhibited that potential Aβ oligomers comprise the major fragments in the hydrophobic central (Aβ$_{13-20}$, HHQKLVFF) and C-terminal (Aβ$_{31-42}$, IIGLMVGGVVIA) regions in high-molecular-weight oligomers (>100 kDa), which clearly identified molecular interactions between Q-OB and Aβ oligomers. Given that the binding of Q-OB is mainly associated with the transient hydrophobic regions on the surface of only in Aβ oligomers intermediates but not in matured fibrils, we

postulate that the interaction of Q-OB and Aβ oligomers is enough to be observed in a biological system. Effective Q-OB labeling of Aβ aggregates was analyzed in the cerebral hippocampal/cortical tissues of AD patients and AD model mice. In vivo imaging successfully visualized Aβ levels with 5xFAD transgenic AD model mice by taking advantage of BBB penetrability and biocompatibility of Q-OB. Eventually, compared to INNOBIA AlzBio3 xMAP ELISA, the Q-OB assay demonstrated a significant correlation with diagnosis using CSF markers of AD patients (i.e., Aβ$_{1-42}$, p-Tau$_{181}$, and t-Tau) along with a change of log ($I/I_0$) values of $0.34 \pm 0.13$ (CN), $0.15 \pm 0.12$ (MCI), and $0.14 \pm 0.10$ (ADD). We believe that our strategy to design an Aβ oligomers-sensing probe has a strong potential to be an excellent starting point for further probe development, such as PET contrast imaging agents by $^{18}F/^{19}F$ isotopic exchange. In addition, these results can advance clinical diagnostic applications for early disease diagnosis and fundamental research on the Aβ oligomer-related pathology of AD.

**Table 1 | Clinical characteristics and CSF biomarkers in AD continuum**

| Multiple analytes | CN[a)] | MCI[b)] | ADD[c)] | *p*-values | Post-hoc. |
|---|---|---|---|---|---|
| Cerebrospinal fluids | (*n* = 22) | (*n* = 20) | (*n* = 19) | | |
| log ($I/I_0$) | 0.34 ± 0.13 | 0.15 ± 0.12 | 0.14 ± 0.10 | *p* < 0.001 | a > b = c |
| $A\beta_{1-42}$ (pg/mL) | 1218 ± 149 | 383 ± 114 | 370 ± 103 | *p* < 0.001 | a > b = c |
| p-Tau$_{181}$ (pg/mL) | 9.84 ± 2.94 | 24.59 ± 9.53 | 22.44 ± 8.22 | *p* < 0.001 | a < b = c |
| NFL (pg/mL) | 576 ± 213 | 737 ± 404 | 985 ± 373 | *p* = 0.001 | a < b < c |
| Neuropsychological domain | (*n* = 22) | (*n* = 20) | (*n* = 15) | | |
| SNSB_Attention (z score) | 0.06 ± 1.20 | −0.12 ± 0.78 | −1.20 ± 1.04 | *p* = 0.002 | a = b > c |
| SNSB_Language (z score) | 0.55 ± 0.49 | −0.01 ± 1.12 | −4.10 ± 5.47 | *p* < 0.001 | a = b > c |
| SNSB_Visuospatial (z score) | 1.04 ± 0.55 | 0.77 ± 0.79 | −2.33 ± 3.62 | *p* < 0.001 | a = b > c |
| SNSB_Memory (z score) | −0.03 ± 0.72 | −1.69 ± 0.87 | −3.05 ± 1.61 | *p* < 0.001 | a > b > c |
| SNSB_Frontal (z score) | 0.17 ± 1.07 | −0.50 ± 1.04 | −3.42 ± 2.81 | *p* < 0.001 | a = b > c |
| Imaging | (*n* = 22) | (*n* = 19) | (*n* = 19) | | |
| SUVR | 0.99 ± 0.05 | 1.40 ± 0.12 | 1.41 ± 0.11 | *p* < 0.001 | a < b = c |

Data are presented as means ± s.d. or numbers (percentages) from individually independent samples. The levels of characteristics and biomarkers were evaluated by analysis of variance with Bonferroni post-hoc test. The exact *p*-values are provided in a Source Data file. The abbreviated words are CN, cognitive normal (healthy control); MCI, mild cognitive impairment; ADD, Alzheimer's disease dementia; $A\beta_{1-42}$, amyloid-$\beta_{1-42}$; p-Tau$_{181}$, phosphorylated tau$_{181}$; NFL, neurofilament light chain; SNSB, the Seoul Neuropsychological Screening Battery; SUVR, standardized uptake value ratio. Source data are provided as a Source Data file.

## Methods

### Inclusion and ethics statements
We confirm that our research complies with all pertinent ethical regulations. All animal procedures were approved by the Committee for the Care and Use of Laboratory Animals at Yonsei University (Korea, IACUC-A-202205-1479-01). The human cerebral hippocampal tissue specimens were received from the Victorian Brain Bank Network (VBBN), supported by The Florey, The Alfred and the Victorian Institute of Forensic Medicine and funded in part by Parkinson's Victoria, MND Victoria and FightMND (Australia). This study was approved by Victorian Institute of Forensic Medicine (VIFM) Ethics Committee and Human Ethics Committee STEMM 1 (Ref. No. 2020-20326-13293-3) of the University of Melbourne. For the clinical study participants, this study included ADD patients and healthy (non-demented) participants who volunteered for a medical examination at the Gwangju Alzheimer's Disease Dementias Cohort Center. The Institutional Review Boards of Chosun University Hospital (CHOSUN 2013-12-018-068), and Chonnam National University Hospital (CNUH-2019-279) provided ethical oversight as part of our observational study of CSF fluorometric analysis. Written informed consent was obtained from each participant or their legal guardian.

### Compound synthesis
The overall synthetic scheme for Q-OB is outlined in Fig. 2a, b. Detailed synthesis methods were described in Supplementary Methods section. All intermediates (compounds 1 and 2) were characterized using [1]H and [13]C nuclear magnetic resonance (NMR) and the final compound was characterized by [1]H, [13]C NMR, and electrospray ionization mass spectroscopy (Supplementary Figs. 1–8). The purity of the compounds was confirmed by high-performance liquid chromatography (Supplementary Fig. 9).

### Synthesis of full-length $A\beta_{1-42}$ and its fragments
$A\beta_{1-42}$ for in vitro assay and a series of Aβ fragment hexamers (1–6 to 37–42) with C terminal cysteine for plate assay were synthesized using solid-phase peptide synthesis. The first C terminal amino acid was coupled with Wang resin LS (0.1 mmol) via symmetric anhydride activation using Fmoc-protected amino acid (0.8 mmol, CEM, USA), diisopropylcarbodiimide (DIC, 0.4 mmol, TCI, Japan) and 4-dimethylaminopyridine (22 mg, Merck, USA) in dimethylformamide (DMF)/dichloromethane (DCM) (1:1, v/v, Samchun, Korea). After 2 h, the resin was washed, and residues were sequentially synthesized using a microwave peptide synthesizer (Liberty blue, CEM, USA).

Following the manual, 1.0 M of Oxyma Pure (CEM, USA) and 1.0 M DIC in DMF as coupling reagents, and 20% piperidine (Alfa-aesar, USA) as deprotecting solution were used. After synthesis, cleavage cocktail (92.5:2.5:2.5:2.5 trifluoroacetic acid (TFA)/DW/triisopropylsilane/3,6-dioxa-1,8-octanedithiol, v/v/v/v) was added to the resin and reacted for 4 h. TFA solution was evaporated with a rotary evaporator and peptide were precipitated using ice-cold diethyl ether. This crude peptide was isolated through centrifugation and dissolved in acetonitrile/DW, followed by lyophilization. Reverse-phase high-performance liquid chromatography (Agilent 1260, Agilent Technologies, USA) was performed to purify peptides. Solvent A (0.1% TFA in DW) and B (0.09% TFA in Acetonitrile) was used for binary gradient. Biphenyl or C18 column (Phenomenex, USA) was used for purification, and products were monitored via a 230 nm UV detector and mass spectrometry (Agilent 6120, Agilent Technologies, USA).

### In vitro Aβ self-assembly kinetics assay
To conduct kinetics assay in separated incubation, prepared 44.3 mL of $A\beta_{1-42}$ dimethyl sulfoxide (DMSO) stock solution (5 mM) was diluted with 4.4 mL PBS (20 mM, pH = 7.4), (working concentration: 50 μM of $A\beta_{1-42}$ containing 1% DMSO). Five sets of diluted solutions were incubated in an orbital shaker (37 °C, 300 rpm); 2970 μL of the incubated solution was collected every 30 min for 6 h and then at 12, 24, 36, 48, 60, and 72 h. Then, 30 μL of Q-OB DMSO solution (100 μM) (or 1 mM of ThT DMSO solution) was added to the collected solution, and the fluorescence spectra were recorded ($\lambda_{ex}$ = 460 nm for Q-OB and $\lambda_{ex}$ = 440 nm for ThT) (slit 2.5/2.5, *n* = 3 independent experiments).

To conduct kinetics assay in coincubation, $A\beta_{1-42}$ and probe (Q-OB or ThT) stock solutions were diluted with PBS (20 mM, pH = 7.4), aCSF (Tocris Bioscience), or DW [working concentration: 10 μM of $A\beta_{1-42}$ and 1 μM of Q-OB (or 10 μM of ThT), respectively, containing 1% DMSO]. The mixture solution (100 μL) was transferred to a 96-well black plate (SPL, Korea) and covered with adhesive optical sealing film (Bioneer Inc., Korea). Fluorescence intensity was measured using a Hidex Sense Microplate Reader every 10 min under orbital-shaking incubation for 12 h (37 °C, 200 rpm) ($\lambda_{ex}$ = 460 nm, $\lambda_{em}$ = 580 nm for Q-OB, and $\lambda_{ex}$ = 440 nm, $\lambda_{em}$ = 490 nm for ThT) (*n* = 3 independent experiments).

### Aβ-species preparation
$A\beta_{1-42}$ monomer: To prepare a stock solution of $A\beta_{1-42}$ (5 mM), 44.3 μL of DMSO was added into the purchased $A\beta_{1-42}$ peptide kit (or 1 mg of

synthesized $A\beta_{1-42}$) and treated with a vortex to be evenly dissolved. Before each experiment, the prepared stock solution was diluted to experimental concentrations with PBS (10 mM, pH = 7.4).

$A\beta_{1-42}$ oligomers: A previously reported procedure was optimized in this work as follows[66]. Prepared DMSO stock solution of $A\beta_{1-42}$ (5 mM) was added to the 4.4 mL of 20 mM PBS (pH = 7.4) and then incubated with an orbital shaker (300 rpm) at 37 °C for 2 h. The incubated solution was mildly pipetted, and the filtrate was collected using polytetrafluoroethylene polymer (PTFE) membrane syringe filters (Whatman GD/X, pore size 0.2 μm, diameter 13 mm). The filtrate was centrifuged (14,000 × g for 10 min at 4 °C), and the supernatant was diluted to experimental concentrations with PBS (10 mM, pH = 7.4) before each experiment.

$A\beta_{1-42}$ fibrils: Prepared DMSO stock solution of $A\beta_{1-42}$ (5 mM) was added to the 4.4 mL of 20 mM PBS (pH = 7.4) and then incubated with an orbital shaker (300 rpm) at 37 °C for 72 h. The incubated solution was mildly pipetted and diluted to experimental concentrations with PBS (10 mM, pH = 7.4) before each experiment.

$A\beta_{1-40}$ monomer: To prepare stock solution of $A\beta_{1-40}$ (5 mM), 46.2 μL of DMSO was added into the purchased $A\beta_{1-40}$ peptide kit and treated with a vortex to be evenly dissolved. Before each experiment, the stock solution was diluted to experimental concentrations with PBS (10 mM, pH = 7.4) before each experiment.

$A\beta_{1-40}$ fibrils: The prepared DMSO stock solution of $A\beta_{1-40}$ (5 mM) was added to the 4.6 mL of 20 mM PBS (pH = 7.4) and then incubated with an orbital shaker (300 rpm) at 37 °C for 168 h. Before each experiment, this incubated solution was mildly pipetted and diluted to experimental concentrations with PBS (10 mM, pH = 7.4).

### Transmission electron microscopy

$A\beta_{1-42}$ monomer (50 μM) incubated at 37 °C with orbital shaking (300 rpm) in PBS (20 mM, pH = 7.4) or aCSF was applied to freshly glow-discharged carbon-coated copper grid at selected time points (0, 2, 6, 12, and 24 h). After allowing the sample to absorb for 2 min and blotting off buffer solution onto Whatman filter paper, then the sample on the grids was stained with Uranyless (EMS) for 1 min. Then it was blotted off Uranyless. Then the grids were examined using a transmission electron microscope (Bio-High voltage EM) (JEM-1400 Plus at 120 kV; JEOL Ltd., Tokyo, Japan) at Korea Basic Science Institute (KBSI) Ochang Center.

### Spectroscopy in the presence of $A\beta_{1-42}$ and potential interferents (proteins, metal ions, amino acids, and thiols)

Proteins, including, $A\beta_{1-42}$, BSA, HSA, and p-Tau, were diluted to 10 μM in PBS (10 mM, pH = 7.4) and metal ions, amino acids, and thiols were dissolved to 1 mM in DW. Stock solutions of Q-OB were prepared in DMSO and added to solutions of each analyte to finally obtain a solution containing 1% DMSO. All fluorescence emission signals were obtained after 10 min incubation at 37 °C with 300 rpm orbital shaking.

### Fitting of $A\beta_{1-42}$ saturation titration analysis

Aβ-species (monomer, oligomers, and fibrils of $A\beta_{1-40}$ or $A\beta_{1-42}$) and probes (Q-OB and ThT) are virtually non-fluorescent, and a simplified Eq. (1) can be used where the fluorescence intensity is directly proportional to the concentration of the Aβ–probe (Q-OB or ThT) complex only:[25]

$$I = \frac{1}{2}F \times \left( \left([P] + N[A\beta] + K_D\right) - \sqrt{\left([P] + N[A\beta] + K_D\right)^2 - 4N[A\beta][P]} \right)$$
(1)

With $F$: a fluorescence proportionality factor, [P]: the initial concentration of Q-OB (or ThT), [Aβ]: the initial concentration of Aβ, N: the number of equivalent binding sites on the Aβ-species relative to Aβ monomers, and $K_D$: the dissociation constant. The results in Fig. 3c and Supplementary Fig. 28 were obtained following a multi-parameter

optimization of the experimental results following the fluorescence intensity at 440 or 460 nm for ThT or Q-OB, respectively, using a concentration of 10 μM Aβ-species and total concentrations of 25 μM for Q-OB or ThT in PBS (10 mM, pH = 7.4) containing 1% DMSO. The slit width was set at 2.5/2.5.

### Limit of detection

Fluorescence intensity of Q-OB (1 μM) was measured with various concentrations (0, 0.125, 0.25, 0.5, 1, 2.5, 5, 7.5, and 10 μM) of $A\beta_{1-42}$-species (monomer, oligomers, and fibrils). Limit of detection was calculated with Eq. (2):

$$LOD = 3\sigma/k$$
(2)

Where σ is the standard deviation of the blank and $k$ is the slope of each plot. ($\lambda_{ex}$ = 460 nm, $\lambda_{em}$ = 580 nm, slit 2.5/2.5, $n$ = 3 independent experiments).

### Fragment mapping assay

Fragment plate screening was performed following a previously reported method[43]. Cysteine-coupled full-length $A\beta_{1-42}$ and hexamer fragments were immobilized on maleimide-activated microplates. Unbound peptides were washed, and excess maleimide residues were deactivated using cysteine solution. To determine Q-OB-interacting Aβ residues, Q-OB stock in DMSO (5 mM) was diluted with a binding buffer (0.1 M sodium phosphate, 0.15 M sodium chloride, and 10 mM EDTA; pH = 7.2) to 5 μM, and then diluted Q-OB solution was added to each well (100 μL per well). The plate was incubated in an orbital-shaking incubator (37 °C, 800 rpm). After discarding Q-OB solution, the plate was washed to remove unbound Q-OB, and the fluorescence was measured at $\lambda_{ex}/\lambda_{em}$ = 500/620 nm using a multiplate reader. To investigate the interaction between Q-OB and fragment-induced Aβ oligomers, the wells were treated with 25 μM of full-length $A\beta_{1-42}$ was treated and incubated for 2 h in a 37 °C orbital-shaking incubator (800 rpm). The plate was washed and incubated with 1 μM Q-OB overnight under the same conditions. The fluorescence signal was similarly measured after washing. The baseline fluorescence intensity of the untreated wells was subtracted from that of each corresponding well of the probe-treated wells. Normalized data were obtained using Eq. 3.

$$\text{Fluorescence intensity}(\%) = \frac{A\beta_{n-(n+5)}(+)\text{probe} - MAP}{A\beta_{1-42}(+)\text{probe} - MAP} \times 100$$
(3)

### Molecular weight cut-off assay

Full-length $A\beta_{1-42}$ was dissolved in DMSO to prepare a 5 mM stock solution, followed by dilution to 20 μM with DW containing 1% DMSO. The solutions were incubated in an incubator at 37 °C for 2 h. A cut-off filter with a molecular weight of 100 kDa was utilized to separate high-ordered aggregates and low-molecular-weight oligomers. For filtration, 200 μL of the sample was loaded and then centrifuged at 14,000 × g for 30 min and then at 1000 × g for 2 min at 4 °C. The filtered and non-filtrated fractions were collected, and the non-filtrated fraction was diluted 10-fold owing to their low volume. The prepared Aβ samples were mixed with equal amounts of 2 μM Q-OB solution in DW, and fluorescence was measured ($\lambda_{ex}/\lambda_{em}$ = 490/620 nm) on a 96-well black plate (Corning, USA) with a microplate reader at regular intervals for 24 h (0.5, 1, 1.5, 2, 4, 12, and 24 h).

### Thioflavin T assay

To examine the aggregation of $A\beta_{1-42}$, a ThT assay was conducted. ThT was dissolved in 50 mM glycine buffer (pH = 8.5). For the assay, 25 μL of the aforementioned $A\beta_{1-42}$ solution was seeded on the 96-half-well black plate, and 75 μL of 5 μM ThT solution was added. After shaking,

fluorescence intensity was measured at $\lambda_{ex}/\lambda_{em} = 450/480$ nm with a microplate reader.

## Preparation of mice brain tissues

Laboratory mice were maintained at macroenvironmental temperature and humidity ranges of 65 to 75 °F (18–23 °C) and 40% to 60%, respectively. Sex was not considered in this study design or methods. To determine whether Q-OB dye could stain Aβ aggregates, 5xFAD transgenic AD model mice presenting major features of AD amyloid pathology, including intraneuronal $Aβ_{1-42}$-induced neurodegeneration and amyloid aggregate formation, were prepared in this study and the B6 wild-type mice were used as controls. For tissue preparation, 12- and 16-month-old 5xFAD transgenic AD model mice and 8-month-old B6 wild-type mice were sacrificed after transcardiac perfusion with saline. Brain tissues were fixed in 4% paraformaldehyde (Biosesang, Korea) for 24 h and immersed in 30% sucrose for 48 h for cryoprotection. Frozen brain tissues were sliced by 25 μm thickness with a cryostat (CM1860, Leica, Germany).

## Staining of 5xFAD and B6 wild-type mice brain tissues

For brain staining, antigens were retrieved with 1% sodium dodecyl sulfate (SDS) (Biosesang, South Korea) in PBS (w/v). DMSO stock of Q-OB (5 mM) was diluted to 0.01, 0.1, 0.5, and 1 μM with 1% DMSO in PBS and the brain tissues were stained with diluted Q-OB solution for 1 h at room temperature. After staining with Q-OB, the tissues were blocked with 20% horse serum (16050130, Thermo-Fisher, USA) in PBS for one h at room temperature. The brain slices were sequentially immunostained with anti-Aβ monoclonal antibody 6E10 (1:200, BioLegend, USA, #SIG-39320) for 2 h and goat anti-mouse IgG conjugated with Alexa 350 (1:200, Invitrogen, USA, #A11045) for 1 h at room temperature.

## Preparation of human brain tissues

Formalin-fixed and paraffin-embedded human postmortem cerebral hippocampal tissue specimens from a total of 13 individuals (AD patient group of four ε3 homozygotes, two ε2/ε3 individuals, and one ε3/ε4 individual and non-patient group of five ε3 homozygotes and one ε2/ε3 individual) were obtained from the Victorian Brain Bank Network (VBBN) without clinical information. The controls are all individuals who have had a full neuropathological examination and have no significant neuropathology. Their clinical history also has no history of significant neurological disease of any type.

## Staining of human brain tissues

Cerebral hippocampal tissues were immersed in xylene three times for 10 min to remove paraffin, followed by two 10 min washes with a series of diluted alcohols (100, 90, 70, and 50% ethanol in DW, v/v). After washing, antigen was retrieved with citrate buffer (10 mM, pH = 6.0) at 95 °C, 30 min. The brain slices were moved to the refrigerator for 1 h and stained with Q-OB and either anti-Aβ monoclonal antibody 6E10 (1:200, BioLegend, USA, #SIG-39320) or 4G8 (1:200, Biolegend, USA, #800701) under the same conditions as the mice brain slides.

## Fluorescence image acquisition of brain tissues

The stained tissues were visualized under a fluorescence microscope (DM500, Leica, USA) equipped with filter cubes: an A4 filter cube to detect 6E10 or 4G8 (excitation filter: BP 360/40; dichromatic mirror: 400; emission filter: BP 470/40), N2.1 filter cube (excitation filter: BP 515–560; dichromatic mirror: 580; emission filter: LP 590), and L5 filter cube to detect Q-OB (excitation filter: BP 480/40; dichromatic mirror: 505; emission filter: BP 527/30).

## In vivo fluorescence imaging

2-month-old wild-type B6 mice ($n = 3$ biologically independent animals), 3-month-old 5xFAD-Tg mice ($n = 3$ biologically independent animals), and 13.5-month-old mice ($n = 3$ biologically independent animals) were housed in a room with 12 h light–dark cycle. The temperature was maintained at 21 °C, and the mice were allowed free access to food and water. For the injection, 5 mM Q-OB was mixed with an equal amount of Tween 80, followed by dilution to 500 μM using PBS. To minimize the interference of mouse fur, the top of the mouse head was shaved before imaging. Shaving was performed after isoflurane inhalation, and 200 μL Q-OB solution was injected intravenously into the mice. The mice were then anesthetized for an additional 1 min and placed on a stage. The ventilator supplied isoflurane gas; stage temperature was maintained at 37 °C. The fluorescence intensity of the cranial region was measured ($\lambda_{ex}/\lambda_{em} = 500/620$ nm) immediately after positioning (1 min time-point). At 5, 10, 20, 35, and 50 min time-points, fluorescence intensity was measured after confirming whether the mice survived.

## Clinical study participants

A total of 61 subjects were examined by neurologists and received a full dementia-screening test, which consisted of laboratory and detailed neuropsychological tests (Seoul Neuropsychological Screening Battery)[65] and brain images with Florbetaben PET. Cognitive impairment was defined as a z-score of less than 1.5 standard deviations (normed for age and education) on at least one of the neuropsychological tests (i.e., a memory, language, visuospatial function, attention, or frontal/executive test). In the design and methodology of this study, no social constructs or relevant grouping were employed. The variable of sex was not a factor considered in the experimental design or analytical methods.

## CSF collection and analysis

CSF samples were obtained between 8 am and 10 am via lumbar puncture with an aseptic technique at the L3-L4 or L4-L5 intervertebral spinous process space using a 22- or 21-gauge needle. CSF was collected in Falcon polypropylene tubes (BD Biosciences, Franklin Lakes, NJ, USA). The first 2–3 mL of CSF was analyzed for routine chemical parameters, including cell count, glucose level, and total protein concentration. Within 15 min, the remaining CSF samples were centrifuged for 10 min at 1000 × g and 4 °C to remove cells. Aliquots (0.45 mL) were added to polypropylene tubes and stored at –80 °C until analysis[67,68]. The concentrations of $Aβ_{1-42}$, p-Tau$_{181}$, and t-Tau in CSF were measured using the INNOBIA AlzBio3 xMAP (INNOTEST) kit with research-only reagents (Fujirebio, Tokyo, Japan, #80584) according to the manufacturer's protocols.

## Q-OB assay for CSF biomarkers

Freeze CSF aliquots were thawed at 0 °C for 30 min, followed at 25 °C for 10 min, and used without a pre-treatment procedure. Prepared $Aβ_{1-42}$ and Q-OB stock solution in DMSO were diluted with CSF (working concentrations: 10 μM $Aβ_{1-42}$ and 1 μM Q-OB in CSF containing 1% DMSO) and treated with a vortex to be evenly dissolved. Then, the Q-OB-treated CSF was transferred to a 96-well black plate (SPL, Korea) (100 μL per well) and covered with an adhesive optical sealing film (Bioneer Inc., Korea). Fluorescence intensity was measured using a microplate reader every 10 min under orbital shaking incubation for 12 h (37 °C, 200 rpm) ($\lambda_{ex}/\lambda_{em} = 460/580$ nm) ($n = 3$ independent experiments).

## Reporting summary

Further information on research design is available in the Nature Portfolio Reporting Summary linked to this article.

# Data availability

The data discussed in this study are presented in the manuscript and supplementary information. Source data are provided with this paper and additional data can be obtained from the corresponding author upon request. Source data are provided with this paper.

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

## Acknowledgements

All the clinical data and CSF samples were provided by the Gwangju Alzheimer's & Related Dementia Cohort Center (Gwangju, Korea). The human cerebral hippocampal tissue specimens were received from the Victorian Brain Bank Network (VBBN), supported by The Florey, The Alfred and the Victorian Institute of Forensic Medicine and funded in part by Parkinson's Victoria, MND Victoria and FightMND (Australia). This research was supported by Basic Science Research Program through the National Research Foundation of Korea (NRF) funded by the Ministry of Education (No. RS-2023-00273030, J.A.), by the Ministry of Health & Welfare and Ministry of Science and ICT (CRI project No. 2018R1A3B1052702, J.S.K.), by the Original Technology Research Program for Brain Science of the NRF, MSIT (No. NRF-2014M3C7A1046041, K.H.L.), by Mid-Career Researcher Program (No. NRF-2021R1A2C2093916, Y.K.), by the Basic Research Program through the Korea Brain Research Institute (KBRI) funded by Ministry of Science and ICT (No. 23-BR-03-05, K.H. L.), and by the Healthcare AI Convergence Research & Development Program through the National IT Industry Promotion Agency of Korea (NIPA) funded by the Ministry of Science and ICT (No. 1711171057, K.H.L.).

## Author contributions

J.A. conceived and designed the research. J.A. performed the molecule design, synthesis, and characterization. J.A. carried out the spectroscopy and in vitro analysis. J.A., K.K., and Y.K. conducted identification of molecular interaction. J.A. and J.S. contributed to DFT calculation. K.K., Hye Yun Kim, and Y.K. carried out the brain tissue imaging. I.W.P., I.C., Hyeong Yun Kim, and S.K. performed in vivo imaging. C.M. contributed to preparation of brain tissue specimens. J.A., H.J.L., K.Y.C., and K.H.L. achieved CSF biomarkers analysis. J.A. and J.S.K. wrote the manuscript. All authors contributed to the discussion and gave approval to the final version of the manuscript.

## Competing interests

The authors declare no competing interests.
