## [Peer Review File · Nature Communications]

Early onset diagnosis in Alzheimer's disease patients via amyloid- β oligomers-sensing probe in cerebrospinal fluidReviewers' Comments:

Reviewer #2:

Remarks to the Author:

The process of A β aggregation has been the subject of numerous studies over the past years, mainly due to its relevant role in the development of Alzheimer's disease (AD). Unfortunately, despite the efforts of numerous research groups, we do not yet have any approach to translate into a reliable diagnostic method for early AD diagnosis. In this context, the study presented by An et al. describes the design and validation of a fluorescent probe that targets A β 42 aggregated species, particularly oA β 42. It uses a nice mixture of fluorescence methods and biochemical and in vivo assays to make their case.

1) Confidently identifying the time point in the aggregation reaction where A β 42 oligomers are most abundant in solution would be very helpful to assess the relationship between the presence of oligomeric species and Q-OB fluorescence intensity changes. mA β 42 and oA β 42 do not bind Th-T. Therefore, comparing A β aggregation kinetics using Q-OB and Th-T seems not to be the best choice, ANS could be more informative in terms of oligomeric species (Fig 2, panel c and d)

2) The lack of data on the characterization of the A β 42 preparations used in this study makes it difficult to evaluate the interpretation of the results and the paper's main claims. Simply citing the previous publications, some of which do not provide the information mentioned, is not sufficient and does not allow one to properly assess the experiments and the interpretation of the results from the experiments performed by the authors.

For instance, Fig 3 electron microscopy would allow the assessment of the morphology of panel d, e, and f preparations.

3) Specificity of the fluorescent probe to A β species:

The claim is made that the fluorescent probe Q-OB is A β species-specific, and while the authors demonstrate that they only bind certain A β aggregates, there is no evidence to suggest specificity to A β . While the A β species compared are not confronted with aggregates formed from other proteins. As discussed by the authors, the significant oA β selectivity vs iA β is attributed to the transient exposition of hydrophobic regions on the surface (Fig 3). Since the exposition of hydrophobic patches is a common feature of other aggregated proteins, this means that the use of Q-OB as a diagnostic system is unclear. Would it also be possible to determine whether Q-OB is also able other oligomeric proteins?

4) In biofluids, such as CSF, the presence of a vast excess of monomers demands high selectivity for oligomers over monomeric species. Thus, a major limitation of oligomer-specific diagnostic methods is the unwanted cross-reaction with the large excess of mA β the fluid. Therefore, the authors should consider performing cross-reactivity measures testing if the presence of high mA β concentrations (e.g. 100 or 200 fold mA β with respect of the oA β concentration) interferes with the binding of Q-OB to oA β .

5) Regarding the use of Q-OB as a potential imaging agent:

- No in vivo stability, pharmacokinetic, or toxicity studies are reported in this manuscript. Therefore, the authors should highlight this in a potential revised manuscript.

- The fluorescent pattern for Young AD and control mice is identical in the data presented (Fig 4 panels b and c). Suggesting that Q-OB may present off-target binding events over plasma proteins and not selectively to A β aggregated species. Moreover, the fact that only for Old AD mice Q-OB seems to possess a certain fluorescence signal suggests that Q-OB may not have the required sensitivity for the early and transient oA β 1 aggregated species.

The measurement of oligomers is technically challenging by the minute amount of oligomers in human biofluids, such as cerebrospinal fluid, requiring extremely sensitive probes. The in vivo data presented here suggests that the technique is not delivering the demanded sensitivity to ensure the limit of

detection required, at least for in vivo imaging. This observation could be related to the revealed binding affinity of Q-OB for oA β of K_d 302.63 ± 94.75 nM. Following this line of thinking, could it be the case that a redesigned variant providing a higher number of Q-OB binding sites (N) per A β may result in an enhancement in the fluorescence signal enough for the low-populated oligomers detection in the early stages of the disease.

6) The authors include a scheme showing the experimental assay for Q-OB clinical application. Exist some indispensable steps for establishing an assay as an efficient diagnostic test with expanded clinical implementation. The assay must display high inter-laboratory reproducibility, should be short, and not require handling by trained staff. Is the methodology presented here easy enough to fulfill these requirements for being implemented in routine AD diagnostics? A discussion of caveats should be included to address this point.

Reviewer #3:

Remarks to the Author:

Comments on manuscript NCOMMS-23-12055 submitted to the Nature Communications

TITLE: "Pioneering the Early Diagnosis of Alzheimer's Disease: A β Oligomers-Sensing Probe for Cerebrospinal Fluid Analysis in Patients"

Authors: Jusung An, Kyeonghwan Kim, Ho Jae Lim, Jinwoo Shin, In Wook Park, Illhwan Cho, Hyeong Yun Kim, Sunghoon Kim, YoungSoo Kim, Kun Ho Lee and Jong Seung Kim

Alzheimer's disease (AD) current clinical diagnosis methods are based on symptomatology and can solely be used when disease is installed and has already progressed extensively. Even so, the accuracy is rather low. So, a methodology that overcomes these difficulties is an obvious clinical need.

Amyloid- β oligomers (oA β) are closely correlated with the synaptotoxicity and neuronal death in the brain, that leads AD and the corresponding cognitive impairment. Since there is a lag between the oA β appearance and either AD symptoms of dementia or clinical diagnosis (e.g. positron emission tomography), the oligomers detection in a very early stage of disease progression is thus of the outmost importance in order to allow a well-timed intervention of treatment.

A highly selective and sensitive fluorescent probe for detecting oA β over monomers (mA β) and insoluble fibrils (iA β), which do not correlate with the AD progression, was synthesized and proposed by the authors. The proof-of-concept relative to the cerebrospinal fluid of AD patients was presented. The proposed methodology allows not only to detect early stages of disease, before symptoms of incipient dementia appear, but also to follow its progression by dynamically monitoring oA β during amyloid fibrillogenesis. The "Q-OB method" was compared with a commercially available platform.

This manuscript is therefore, in my opinion, suitable for publishing in Nature Communications, after some minor revision:

1. In the introduction the writing must be uniform, as there are some parts that are not intelligible (mainly in the 2nd and 3rd paragraphs). Moreover, sentences like "Small-molecule-based fluorescence spectroscopy is the most practical and affordable technology" must be avoided, since they are unproven and highly questionable.
2. The CMC value of $28.8 \mu\text{M}$ does not agree with Figure S13a which shows $28.872 \mu\text{M}$, so $28.9 \mu\text{M}$ should be the correct value.
3. Where is Table 5 referred to in page 15?
4. The variation in fluorescence intensity of Q-OB in the patients' CSF, between the values obtained for "mild cognitive impairment" and "AD dementia" is quite small (from 0.15 to 0.14). However, the authors state that it is possible to "distinguish an AD continuum state". Authors should clarify this claim.
5. The correlations between the ATN CSF biomarkers concentrations and fluorescence intensity values are far from linear ($R^2 < 0.36$) (Figures 5c-e). The authors should correct this statement or comment on this issue.

Materials and Methods are consistently described. Some minor remarks:

1. On page 4 of Supplementary Information (SI) the statement "ωB97XD functional at the N07D level of theory" is not correct. The N07D is a basis set, not a "level of theory". At most "ωB97XD/N07D level of theory".
2. On page 2 (SI) the experimental conditions for the NMR experiments are presented: "a 500MHz Bruker NMR spectrometer" but the ¹³C spectra are recorded at 125 MHz (Figures S2, S4 and S6).

Reviewer #4:

Remarks to the Author:

This is an interesting study but the CSF Abeta oligomer data should be compared with other available methods to measure Abeta oligomers in CSF, e.g., homogeneous ELISA (the same capture and detection antibody against an N-terminal Abeta epitope). It has been shown before that AD patients have slightly increased AbetaO concentrations in CSF. How does this new method compare to these older methods?

Answer to referees:

Reviewer #1:

The process of A β aggregation has been the subject of numerous studies over the past years, mainly due to its relevant role in the development of Alzheimer's disease (AD). Unfortunately, despite the efforts of numerous research groups, we do not yet have any approach to translate into a reliable diagnostic method for early AD diagnosis. In this context, the study presented by An et al. describes the design and validation of a fluorescent probe that targets A β 42 aggregated species, particularly oA β 42. It uses a nice mixture of fluorescence methods and biochemical and in vivo assays to make their case.

Query 1. Confidently identifying the time point in the aggregation reaction where A β 42 oligomers are most abundant in solution would be very helpful to assess the relationship between the presence of oligomeric species and Q-OB fluorescence intensity changes. mA β 42 and oA β 42 do not bind Th-T.

Therefore, comparing A β aggregation kinetics using Q-OB and Th-T seems not to be the best choice, ANS could be more informative in terms of oligomeric species (Fig 2, panel c and d).

Response: We appreciate your valuable feedback on improving the quality of our manuscript. The fluorescent probe 1-anilino-8-naphthalene sulfonate (ANS) has been widely recognized as a fluorogenic sensor for amyloid- β (A β) fibrillogenesis *in vitro*. Since thioflavin T (ThT) cannot directly assess A β oligomers (oA β) as a reference dye in our proposed experiments, we conducted additional measurements of absorption and emission spectral changes of both fluorescent dyes (**Q-OB** and ANS) under various incubation conditions using phosphate-buffered saline (PBS) (20 mM, pH = 7.4), artificial cerebrospinal fluid (aCSF), and deionized water (DW). Our photo-spectrometry studies suggested that ANS exhibits no selectivity among A β monomer (mA β), oA β , and A β fibrils (iA β) in PBS, as depicted in Figure 1 below (**Figure S18** in the revised supplementary information). In contrast, we observed that **Q-OB** shows a selectivity towards oA β among various A β -species in PBS, as shown in Figure 2 below (**Figure S16** in the revised supplementary information). As expected, aCSF condition, mimicking the cerebrospinal fluids (CSF) microenvironment, led to the different fluorescence signals of **Q-OB** towards various A β -species in contrast to those of PBS. As a negative control, DW condition demonstrated the nonapparent A β fibrillogenesis in **Q-OB** and ThT fluorescence response (Figure 3 below, added to the revised supplementary information as **Figure S17**).

Furthermore, a coincubation kinetic assay experiment using **Q-OB**, ANS, and ThT in PBS, aCSF, and DW media indicated that ANS exhibits a non-specific response to certain A β -species compared to **Q-OB**, or ThT as demonstrated in Figure 4 below (**Figure S19** in the revised supplementary information). Our own experimental findings confirmed that ANS is not selective for oA β , which are consistent with previous reports (DOI: 10.1080/19336896.2020.1720487; 10.1016/j.bbapap.2007.01.002). A detailed discussion, along with additional experiments, has been included in the revised manuscript.

Figure 1. Absorbance and fluorescence spectroscopy of 8-anilino-naphthalene-1-sulfonic acid (ANS). Absorption spectra of ANS (10 μM) in the presence of amyloid- β ($\text{A}\beta$)₁₋₄₂ (10 μM) at various time points (0, 2, and 24 h) of incubation in (a) phosphate-buffered saline (PBS) (20 mM, pH = 7.4), (b) artificial cerebrospinal fluid (aCSF), and (c) deionized water (DW). Fluorescence spectra for identifying $\text{A}\beta$ ₁₋₄₂-species (10 μM) using ANS (10 μM) at various time points (0, 2, and 24 h) of incubation in (d) PBS, (e) aCSF, and (f) DW (slit 2.5/2.5).

Figure 2. Absorbance and fluorescence spectroscopy of Q-OB. Absorption spectra of Q-OB (5 μM) in the presence of $\text{A}\beta$ ₁₋₄₂ (10 μM) at various time points (0, 2, and 24 h) of incubation in (a) PBS (20 mM, pH = 7.4), (b) aCSF, and (c) DW. Fluorescence spectra for identifying $\text{A}\beta$ ₁₋₄₂-species (10 μM) using Q-OB (1 μM) at various time points (0, 2, and 24 h) of incubation in (d) PBS, (e) aCSF, and (f) DW (slit 2.5/2.5).

Figure 3. Absorbance and fluorescence spectroscopy of Thioflavin T (ThT). Absorption spectra of ThT (10 μ M) in the presence of A β ₁₋₄₂ (10 μ M) at various time points (0, 2, and 24 h) of incubation in (a) PBS (20 mM, pH = 7.4), (b) aCSF, and (c) DW. Fluorescence spectra for identifying A β -species (10 μ M) using ThT (10 μ M) at various time points (0, 2, and 24 h) of incubation in (d) PBS, (e) aCSF, and (f) DW (slit 2.5/2.5).

Figure 4. a) Identification of A β ₁₋₄₂ aggregation kinetics using Q-OB (1 μ M), ThT (10 μ M), and ANS (10 μ M) under the coinubation at 37°C for 24 h with A β ₁₋₄₂ (10 μ M) in PBS (20 mM, pH = 7.4) and (b) normalized plot obtained from (a). A β ₁₋₄₂ aggregation kinetics using Q-OB, ThT, and ANS under the coinubation at 37°C for 24 h with A β ₁₋₄₂ (10 μ M) in (c) aCSF and (d) DW. Error ranges represent standard deviation (SD) ($n = 3$).

Query 2. The lack of data on the characterization of the A β 42 preparations used in this study makes it difficult to evaluate the interpretation of the results and the paper's main claims. Simply citing the previous publications, some of which do not provide the information mentioned, is not sufficient and does not allow one to properly assess the experiments and the interpretation of the results from the experiments performed by the authors.

For instance, Fig 3 electron microscopy would allow the assessment of the morphology of panel d, e, and f preparations.

Response: We appreciate your valuable comments. As per your request, we conducted careful biophysical characterization of A β ₁₋₄₂ using bio-transmission electron microscopy (Bio-TEM) on Bio-High voltage EM (JEM-1400 Plus at 120 kV; JEOL Ltd., Tokyo, Japan) during its fibrillogenesis. The results are presented in Figure 5 below, which has been also added to the revised manuscript and supplementary information as **Figures 2g** and **S22**. We obtained clear evidence of the morphology of A β ₁₋₄₂-species, including prefibrillar *o*A β ₁₋₄₂ and *i*A β ₁₋₄₂, for 24 h incubation in PBS (20 mM, pH = 7.4). We also confirmed that after prefibrillar *o*A β ₁₋₄₂ aggregations were formed approximately for 6–12 h, then the matured fibrils were produced within 24 h.

Furthermore, we investigated the difference between incubation conditions using PBS and aCSF by examining the morphology of A β ₁₋₄₂-species in aCSF. Interestingly, after 2 h of incubation, prefibrillar *o*A β ₁₋₄₂ had different shapes as short fibers and spheres in PBS and aCSF, respectively. Both conditions induced an entangled form of oligomer agglomeration after approximately 6 h of incubation. Subsequently, the oligomer agglomeration typically matured into fibrils in PBS condition, while we observed that gradually enlarged oligomer agglomeration occurred only in the aCSF condition. Therefore, we believe that these TEM results not only accurately sampled A β but also explained why **Q-OB** exhibited significantly different photophysical properties with various A β -species in aCSF (or CSF) compared to those in PBS. We have added the data and discussions to the revised manuscript.

Figure 5. Biophysical characterization using bio-transmission electron microscopy (Bio-TEM) images of A β ₁₋₄₂ at specific time points (0, 2, 6, 12, and 24 h) of fibrillogenesis in PBS (20 mM, pH = 7.4) and aCSF.

Query 3. Specificity of the fluorescent probe to A β species:

The claim is made that the fluorescent probe Q-OB is A β species-specific, and while the authors demonstrate that they only bind certain A β aggregates, there is no evidence to suggest specificity to A β . While the A β species compared are not confronted with aggregates formed from other proteins. As discussed by the authors, the significant oA β selectivity vs iA β is attributed to the transient exposition of hydrophobic regions on the surface (Fig 3). Since the exposition of hydrophobic patches is a common feature of other aggregated proteins, this means that the use of Q-OB as a diagnostic system is unclear. Would it also be possible to determine whether Q-OB is also able other oligomeric proteins?

Response: Thanks for your valuable feedback on this point. Explaining the selectivity of **Q-OB** towards oA β is a significant concern. The mapping assay discussed in **Figure 3** (main text) and **Figure S28** (Supplementary information) estimated that **Q-OB**'s specificity to oA β is due to the exposure of temporary hydrophobic regions, especially the hydrophobic (14–21) and C-terminal (29–42) regions of A β _{1–42} peptide. Although hydrophobic patches are common features of other amyloid proteins, including A β _{1–40}, **Q-OB**'s specificity was confidently evaluated for A β _{1–42}. To provide additional evidence, we performed a co-incubated kinetic assay using **Q-OB** with certain amyloid proteins (A β _{1–40} and phosphorylated tau (p-Tau)), as well as albumins (human serum albumin (HSA) and bovine serum albumin (BSA)) as typical interferants for a hydrophobic fluorescent dye. As expected, those interferants negligibly influenced prefibrillar oA β _{1–42} sensing with **Q-OB** (Figure 6 below, **Figure S27** in the revised supplementary information).

Figure 6. Identification of A β _{1–42} aggregation kinetics using **Q-OB** (1 μ M) in the presence of A β _{1–42} (10 μ M) in PBS (20 mM, pH = 7.4) in comparison with potential interferants including (a) A β _{1–40} (10 μ M), (b) phosphorylated tau (p-Tau) (10 μ M), (c) human serum albumin (HSA) (40 mg/mL), and (d) bovine serum albumin (BSA) (40 mg/mL). Error ranges represent SD ($n = 3$).

The marginal increment of fluorescence intensity of **Q-OB** towards analyte mixture of $A\beta_{1-40}$ and $A\beta_{1-42}$ might be attributed to signal amplification by promoting oligomerization of $A\beta_{1-40}$ by $A\beta_{1-42}$ (DOI:10.1038/aps.2017.28). p-Tau demonstrated negligible influence on the fluorescence response of **Q-OB**. Furthermore, albumins induced an earlier peak of fluorescence intensity for the oligomer-induced signal of **Q-OB**, approximately 20 min. Overall, the fluorescence change of **Q-OB** is entirely independent of those potential interferants, confirming again that **Q-OB** is exclusively responsive to $oA\beta_{1-42}$ over other competitive amyloid proteins. These discussions have been added to the revised manuscript.

Query 4. In biofluids, such as CSF, the presence of a vast excess of monomers demands high selectivity for oligomers over monomeric species. Thus, a major limitation of oligomer-specific diagnostic methods is the unwanted cross-reaction with the large excess of $mA\beta$ the fluid. Therefore, the authors should consider performing cross-reactivity measures testing if the presence of high $mA\beta$ concentrations (e.g. 100 or 200 fold $mA\beta$ with respect of the $oA\beta$ concentration) interferes with the binding of **Q-OB** to $oA\beta$.

Response: Thanks for pointing out the considerable issue. We carefully designed an additional *in vitro* experiment to examine the cross-reactivity of **Q-OB** toward various $A\beta$ -species to confirm its $oA\beta_{1-42}$ selectivity. The fluorescence intensity of **Q-OB** (1 μM) was recorded with $oA\beta_{1-42}$ (0.1 μM) in the presence of cross-reactive interferents, including $mA\beta_{1-40}$, $mA\beta_{1-42}$, $iA\beta_{1-40}$, or $iA\beta_{1-42}$ prepared in various concentration ratios of $oA\beta_{1-42}$ ($\times 0$ (blank), $\times 1$, $\times 10$, $\times 100$, and $\times 500$) in PBS (20 mM, pH = 7.4). In each experimental condition, we found that the fluorescence intensity of **Q-OB** did exclusively changes with only $oA\beta_{1-42}$ while negligibly changed in the presence of cross-reactive interferents. **Q-OB** was not cross-reactive to even more than 100-times concentrated $A\beta_{1-40}$ species (monomer and insoluble fibrils). **Q-OB** was found to be cross-reactive to both $A\beta_{1-42}$ species (monomer and insoluble fibrils) with a concentration of 10-times or less, as seen in Figure 7 below (added to the revised supplementary information as **Figure S26**).

Figure 7. Identification of cross-reactivity of **Q-OB** for A β -species. Fluorescence emission intensity of **Q-OB** (1 μ M) with oA β ₁₋₄₂ (0.1 μ M) in PBS (10 mM, pH = 7.4) containing 1% DMSO in comparison with varying concentrations of cross-reactive interferent, including (a) mA β ₁₋₄₀, (b) iA β ₁₋₄₀, (c) mA β ₁₋₄₂, and (d) iA β ₁₋₄₂. Error ranges represent SD ($n = 3$), slit 2.5/2.5.

However, it should be noted that the cross-reactivities of **Q-OB** toward other A β -species are not significant in our *in vitro* assay for Alzheimer's disease (AD) diagnosis using CSF samples. The human CSF we used for diagnosis contain similar or fewer cross-reactive analytes than A β ₁₋₄₂, as shown in **Table S6** (Supplementary information). Additionally, we observed from the fluorescence changes that the monomeric form of A β ₁₋₄₂ smoothly changes into the oligomer form over time. Therefore, although the monomeric form in the sample is highly concentrated, its cross-reactivity at the time of measurement would be negligible. These discussions have been added to the revised manuscript.

Query 5. Regarding the use of Q-OB as a potential imaging agent:

- No *in vivo* stability, pharmacokinetic, or toxicity studies are reported in this manuscript. Therefore, the authors should highlight this in a potential revised manuscript.

- The fluorescent pattern for Young AD and control mice is identical in the data presented (Fig 4 panels b and c). Suggesting that Q-OB may present off-target binding events over plasma proteins and not selectively to A β aggregated species. Moreover, the fact that only for Old AD mice Q-OB seems to possess a certain fluorescence signal suggests that Q-OB may not have the required sensitivity for the early and transient oA β 1 aggregated species.

The measurement of oligomers is technically challenging by the minute amount of oligomers in human biofluids, such as cerebrospinal fluid, requiring extremely sensitive probes. The *in vivo* data presented here suggests that the technique is not delivering the demanded sensitivity to ensure the limit of detection required, at least for *in vivo* imaging. This observation could be related to the revealed binding affinity of Q-OB for oA β of $K_d 302.63 \pm 94.75$ nM. Following this line of thinking, could it be the case that a redesigned variant providing a higher number of Q-OB binding sites (N) per A β may result in an enhancement in the fluorescence signal enough for the low-populated oligomers detection in the early stages of the disease.

Response: Thank you for bringing up the important issue of applying **Q-OB** for optical bioimaging. We carefully conducted *in vitro* assays to demonstrate the cytotoxicity of **Q-OB**. Fortunately, we found that **Q-OB** did not exhibit any cytotoxicity/phototoxicity at concentrations ranging of 0–100 μ M with 24 h incubation (see Figure 8 below, added to the revised supplementary information as **Figure S29**). Moreover, for phototoxicity of **Q-OB**, we observed that **Q-OB** did not produce reactive oxygen species (ROS) under photoirradiation (see Figure 9 below, added to the revised supplementary information as **Figure S30**). Additionally, we could not externally observe any abnormalities in the organs and whole body of mice during organ distribution and brain imaging after the treatment of **Q-OB** *ex vivo* and *in vivo*.

Figure 8. Cytotoxicity of **Q-OB** *in vitro*. Cell viabilities of SH-SY5Y cell treated with **Q-OB** (0, 0.1, 1, 5, 10, 25, 50, and 100 μM) under photoirradiation/non-irradiation conditions. The cell viability is presented as mean \pm SD ($n = 3$).

Figure 9. Characterization of phototoxicity of **Q-OB**. a) Absorption spectra of 1,3-diphenylisobenzofuran (DPBF) (50 μM) and **Q-OB** (5 μM), and irradiation time-dependent UV/Vis spectrum change of DPBF in the presence of **Q-OB** in DMSO under light irradiation of Xe-lamp up to 5 min ($\lambda_{\text{ex}} = 488$ nm, 1 mW/cm²). b) Absorbance changes of DPBF at 417 nm using **Q-OB**, indocyanine green (ICG), methylene blue (MB), and rose bengal (RB) upon photoirradiation ($\lambda_{\text{ex}} = 488$ (**Q-OB**), 530 (RB), 660 (MB), 800 (ICG) nm, respectively) (photosensitizers conc. = 5 μM).

In addition, as mentioned, it is a significant challenge to differentiate between prodromal (young) AD and older transgenic AD mouse models using optical imaging to target $\alpha\text{A}\beta$. While claiming selectivity for oligomers *in vivo* through optical imaging data is illogical, we would like to point out that $\text{A}\beta$ production occurs after 2 months of age in the 5xFAD transgenic mouse model as mentioned in the manuscript. These facts demonstrate pathological differences between neurodegenerative AD-caused normal aging and transgenic AD models. In general, degenerative AD caused by the failure of $\text{A}\beta$ clearance is expected to have varying degrees of certain $\text{A}\beta$ presence depending on age. For example, in the early stages of AD, the expression of monomers or oligomers increases drastically, and then matured fibrils are gradually deposited along with disease progression. On the other hand, in the transgenic AD model, the production of $\text{A}\beta$ continues to be expressed through age (after 1.5 months old), and the overall expression of $\text{A}\beta$ increases

throughout all age periods (DOI: 10.1523/JNEUROSCI.1202-06.2006). In other words, it is difficult to discuss selectivity due to limiting a specific A β state in such a physiological environment, especially for oligomers.

To demonstrate sensitivity of **Q-OB** to *o*A β , we measured limit of detection (LOD). The LOD was found to be 0.57 ± 0.04 , 25.27 ± 1.94 , and 26.82 ± 0.92 nM for *o*A β_{1-42} , *m*A β_{1-42} , and *i*A β_{1-42} , respectively. This is reasonable to give a large difference in the dissociation constant (K_d) and fluorescence turn-on ratio of **Q-OB** for *o*A β_{1-42} (Figure 10 below, added as an inset to **Figure 2j** in the revised manuscript). The revised manuscript now includes the data and discussions regarding this observation. Detecting oligomers with sensitivity is a significant challenge, especially in biofluids (e.g., plasma or CSF), and it is even more challenging when using small-molecule dyes instead of antibodies. Thus, we appreciate your suggestion that a redesigned variant of **Q-OB** could offer a higher number of binding sites (N) per A β . Your idea has inspired us to consider strategies such as modifying the probe molecular size (e.g., shortening/elongating the π -conjugation) to enhance the sensitivity and selectivity of **Q-OB** towards *o*A β .

Figure 10. Linear correlation between fluorescence intensity of **Q-OB** (1 μ M) at various concentrations (0, 0.125, 0.25, 0.5, 1, 2.5, 5, 7.5, and 10 μ M) of *m*A β_{1-42} , *o*A β_{1-42} , and *i*A β_{1-42} in PBS (10 mM, pH= 7.4) containing 1% DMSO. Error ranges represent SD ($n = 3$), slit = 2.5/2.5.

By the way, we wanted to emphasize in the main text that a major potential application of **Q-OB** is an *in vitro* kinetic assay for early diagnosis of AD using patients' CSF, not for its optical bioimaging. In the current manuscript, we briefly suggested the possibility of *in vivo* application to measure the amount of A β in brain regions using transgenic AD-mouse models. However, further examination, such as pharmacokinetics and *in vivo* toxicity (H&E staining), will be required for *in vivo* application, including fluorescence imaging or positron emission tomography (PET) in future studies. To address the reviewer's comment, we have highlighted these issues in the revised manuscript by stating: "Nevertheless, the results of this optical bioimaging should be interpreted with some level of caution, as it is still constrained for *in vivo* assay. It is still challenging to differentiate between prodromal (early) AD and older transgenic AD-models mice using optical imaging to target *o*A β due to limiting a specific A β state in such a physiological environment, especially for oligomers. Besides, the probe for further optical bioimaging should meet several additional requirements, including pharmacokinetics (logical washout kinetics) and *in vivo* toxicity

(low physiological toxicity). Therefore, these results briefly suggested the possibility of *in vivo* application of **Q-OB** to measure the amount of A β -species in the brain regions using transgenic AD-model live mice.” We hope that the reviewer and potential readers will understand that our main concern in the current studies is not the application of **Q-OB** for optical bioimaging.

Query 6. The authors include a scheme showing the experimental assay for Q-OB clinical application. Exist some indispensable steps for establishing an assay as an efficient diagnostic test with expanded clinical implementation. The assay must display high inter-laboratory reproducibility, should be short, and not require handling by trained staff. Is the methodology presented here easy enough to fulfill these requirements for being implemented in routine AD diagnostics? A discussion of caveats should be included to address this point.

Response: Thank you for your helpful feedback. Currently, establishing the **Q-OB** assay as a standard diagnostic tool for AD is challenging. The current technology requires an experienced clinical pathologist to handle it, and patient sample collection (e.g., CSF) must be carried out through a hospital. Therefore, we have been working on a collaboration with MDs in hospitals to overcome these limitations. This is our next upcoming project that we want to tackle. To address this point, we have revised a sentence in the manuscript as follows: “Collectively, these results suggest that the concentration of the intrinsic amyloid CSF biomarker affects the A β fibrillogenetic kinetics in the **Q-OB** fluorescence response. Therefore, it is noteworthy that the **Q-OB** assay can be a useful tool for analyzing ATN biomarkers in CSF to develop accurate and useful methods for early AD diagnosis. Further, we believe the development of an easy-to-use diagnosis kit would help overcome limitations associated with conventional AD diagnosis methods, such as cognitive assessment questionnaires, functional behavioral assessments, and PET imaging (Figure 5f). Our next study plans involve utilizing **Q-OB** analogues as novel diagnostic platforms for clinical applications, satisfying high inter-laboratory reproducibility, short operating time, and ease of use for a layperson, to create an efficient diagnostic assay with expanded clinical implementation as a standard AD early diagnostic method.”

Reviewer #2:

Comments on manuscript NCOMMS-23-12055 submitted to the Nature Communications

TITLE: “Pioneering the Early Diagnosis of Alzheimer’s Disease: A β Oligomers-Sensing Probe for Cerebrospinal Fluid Analysis in Patients”

Authors: Jusung An, Kyeonghwan Kim, Ho Jae Lim, Jinwoo Shin, In Wook Park, Illhwan Cho, Hyeong Yun Kim, Sunghoon Kim, YoungSoo Kim, Kun Ho Lee and Jong Seung Kim

Alzheimer’s disease (AD) current clinical diagnosis methods are based on symptomatology and can solely be used when disease is installed and has already progressed extensively. Even so, the accuracy is rather low. So, a methodology that overcomes these difficulties is an obvious clinical need.

Amyloid- β oligomers (oA β) are closely correlated with the synaptotoxicity and neuronal death in the brain, that leads AD and the corresponding cognitive impairment. Since there is a lag between the oA β appearance and either AD symptoms of dementia or clinical diagnosis (e.g. positron emission tomography), the oligomers detection in a very early stage of disease progression is thus

of the utmost importance in order to allow a well-timed intervention of treatment.

A highly selective and sensitive fluorescent probe for detecting oA β over monomers (mA β) and insoluble fibrils (iA β), which do not correlate with the AD progression, was synthesized and proposed by the authors. The proof-of-concept relative to the cerebrospinal fluid of AD patients was presented. The proposed methodology allows not only to detect early stages of disease, before symptoms of incipient dementia appear, but also to follow its progression by dynamically monitoring oA β during amyloid fibrillogenesis. The “Q-OB method” was compared with a commercially available platform.

This manuscript is therefore, in my opinion, suitable for publishing in Nature Communications, after some minor revision:

Query 1. In the introduction the writing must be uniform, as there are some parts that are not intelligible (mainly in the 2nd and 3rd paragraphs). Moreover, sentences like “Small-molecule-based fluorescence spectroscopy is the most practical and affordable technology” must be avoided, since they are unproven and highly questionable.

Response: We would like to express our gratitude for carefully reading our manuscript and providing us with helpful feedback. As per your comments, we have revised the introduction, particularly the second and third paragraphs, to make it more understandable in the revised manuscript. Additionally, we have changed the sentence “Small-molecule-based fluorescence spectroscopy is the most practical and affordable technology” to “Small-molecule-based fluorescence spectroscopy is one of the most practical and affordable technologies” to avoid any controversy.

Query 2. The CMC value of 28.8 μM does not agree with Figure S13a which shows 28.872 μM , so 28.9 μM should be the correct value.

Response: Thanks for pointing out the issue. We made a mistake, then revised the CMC value to 28.9 μM in the revised manuscript.

Query 3. Where is Table 5 referred to in page 15?

Response: Thank you for pointing out our shameful mistake. It has been deleted in the revised manuscript.

Query 4. The variation in fluorescence intensity of Q-OB in the patients' CSF, between the values obtained for “mild cognitive impairment” and “AD dementia” is quite small (from 0.15 to 0.14). However, the authors state that it is possible to “distinguish an AD continuum state”. Authors should clarify this claim.

Response: Thanks for your valuable advice. In our manuscript, we aimed to describe how to distinguish between cognitive normal (CN) and mild cognitive impairment (MCI)/AD dementia (ADD). We agree that it is difficult and illogical to differentiate MCI from ADD based on a small gap of only 0.15 to 0.14 in the $\log(I/I_0)$ value. To avoid confusion, we have added the following statement to the revised manuscript: “Based on our experimental data showing a negligible

difference in $\log(I/I_0)$ values between MCI and ADD, it was not feasible to distinguish MCI from ADD. However, there was a marked difference between CN and other two pathological states (MCI and ADD), which suggests that our method can diagnose the pathological progression from CN to MCI in clinical applications.”

Query 5. The correlations between the ATN CSF biomarkers concentrations and fluorescence intensity values are far from linear ($R^2 < 0.36$) (Figures 5c-e). The authors should correct this statement or comment on this issue.

Response: Thanks for pointing out this issue. We amended the sentence in the revised manuscript as follows: “In addition, we have used scatter plots to evaluate the correlation between concentrations of ATN CSF biomarkers and $\log(I/I_0)$ values. For the CSF $A\beta_{42}$, the plot showed a proportional correlation, while CSF p-Tau₁₈₁ and NFL exhibited an inversely proportional correlation among the diagnostic groups (Figures 5c–e).”

Materials and Methods are consistently described. Some minor remarks:

Query 6. On page 4 of Supplementary Information (SI) the statement “ ω B97XD functional at the N07D level of theory” is not correct. The N07D is a basis set, not a “level of theory”. At most “ ω B97XD/N07D level of theory”.

Response: Thanks for raising this issue. We have corrected the sentence in the revised supplementary information as follows: “Structure optimization of **Q-OB** was performed using the Gaussian 16 software package, at the ω B97XD/N07D level of theory using the default integral equation formalism variant of the polarizable continuum model (IEFPCM) solvation model of acetonitrile.”

Query 7. On page 2 (SI) the experimental conditions for the NMR experiments are presented: “a 500MHz Bruker NMR spectrometer” but the ¹³C spectra are recorded at 125 MHz (Figures S2, S4 and S6).

Response: Thanks for your comments. Due to different gyromagnetic ratios between ¹³C and ¹H, the ratio of its magnetic moment to its angular momentum is different. Each nuclear magnetic resonance (NMR) active nucleus has a specific frequency because its gyromagnetic ratio varies. It is well-known that carbon atom has a gyromagnetic ratio of about one-quarter of hydrogen. Thus, at the same magnetic field strength (500 MHz in our case), ¹³C and ¹H have different Larmor frequencies. In more detail, the relationship between field and frequency is shown mathematically by equation (1).

$$dE = hv = h(B_o - B_e)\gamma \quad (1)$$

dE is the energy difference between spin states, and h indicates Planck’s constant. ν is the frequency of the B_1 field, and B_e is a small magnetic field generated by the circulation of electrons of the molecule. B_o is the strength of the external homogeneous magnetic field. γ (gyromagnetic ratio) is a constant property of the particular nucleus. The gyromagnetic ratio for the ¹³C nucleus is 67.262 units, whereas for ¹H is 267.513 (i.e., approximately 4 times). Therefore, ¹H NMR measurement at 500 MHz is equivalent to ¹³C NMR at 125 MHz.

Reviewer #3:

This is an interesting study but the CSF Abeta oligomer data should be compared with other available methods to measure Abeta oligomers in CSF, e.g., homogeneous ELISA (the same capture and detection antibody against an N-terminal Abeta epitope). It has been shown before that AD patients have slightly increased AbetaO concentrations in CSF. How does this new method compare to these older methods?

Response: We greatly appreciate your informative review to raise the quality of our manuscript. As you mentioned, some ELISA methods can quantify $\alpha\text{A}\beta_{1-42}$ *in vitro*, specifically those between dimer and dodecamer in size of oligomers. However, to the best of our knowledge, no ELISA method can quantitatively analyze prefibrillar $\alpha\text{A}\beta$ larger than 100 kDa in CSF. We have a newly discovered kinetic assay using **Q-OB** that is capable of recognizing prefibrillar $\alpha\text{A}\beta$ with a size greater than 100 kDa *in vitro*. Therefore, we are unable to directly compare the two different technologies (ELISA and our system) for quantification of the $\alpha\text{A}\beta$ in CSF you have recommended. However, we appreciate your suggestion and plan to design a novel ELISA system that can quantitatively analyze prefibrillar $\alpha\text{A}\beta$ in biofluid environments for our future research.

Reviewers' Comments:

Reviewer #1:

Remarks to the Author:

I genuinely appreciate the authors' commendable efforts in addressing my queries through rigorous experimentation. Their commitment has undoubtedly resulted in a more robust and captivating manuscript.

Reviewer #2:

Remarks to the Author:

Comments on manuscript NCOMMS-23-12055 submitted to the Nature Communications (Round 2)
TITLE: "Pioneering the Early Diagnosis of Alzheimer's Disease: A β Oligomers-Sensing Probe for Cerebrospinal Fluid Analysis in Patients"

Authors: Jusung An, Kyeonghwan Kim, Ho Jae Lim, Jinwoo Shin, In Wook Park, Illhwan Cho, Hyeong Yun Kim, Sunghoon Kim, YoungSoo Kim, Kun Ho Lee and Jong Seung Kim

In my opinion the answers given by the authors to the questions raised are convincing and the changes introduced as requested.

Therefore, I believe that the article deserves publication in Nature Communications.

Reviewer #4:

Remarks to the Author:

The paper by Kim and co-workers presents the synthesis of a novel ligand for fluorescent detection of one of the pathological entities, amyloid-beta oligomers, seen in Alzheimer's disease.

The design and synthetic routes for generating the ligand, Q-OB, as well as the photophysical characteristics are well-presented. In addition, the ligands selectivity towards amyloid-beta oligomers is well characterized in in vitro fibrillation assays with recombinant A-beta species and the authors are also investigating the ligands performance for optical in vivo imaging in transgenic mouse models, as well as for detection of amyloid-beta oligomers in cerebrospinal fluid samples from patients.

However, to conclude that the performance of Q-OB is restricted to detection of amyloid-beta oligomers, as well as suitable for optical in vivo imaging, further experiments are essential.

In general, the specificity of the dyes towards amyloid-beta oligomers needs to be verified in tissue samples. I would therefore recommend this paper for publication in Nature Communications after the revisions outlined below.

1. The characterization of the ligand, Q-OB, for in vivo imaging, is not optimal. The authors only show macroscopic images (Figure 4 and Supplementary Figures 33 and 34) and images showing the characterization of Q-OB towards different aggregated A-beta species in tissue sections would be essential. In this regard, I suggested that histological staining of brain tissue sections from 5xFAD transgenic mice (young and old) and B6 wild type controls with the ligand, Q-OB, and antibodies towards A-beta should be performed. Is Q-OB staining the ordinary aggregated A-beta pathologies, such as CAA and parenchymal plaques? Is it possible to see oligomeric A-beta? Is there any unspecific staining of the white matter? A figure with zoom in images of the respective pathological entities would be preferable.

2. Since a variety of studies, including high resolution structures obtained by cryo electron microscopy, have shown that the structure of in vitro generated aggregated A-beta species, as well as mouse derived A-beta aggregates, are different from aggregated A-beta pathology derived from human AD

tissue sections, histological staining of brain tissue sections from human AD cases and age matched controls with the ligand, Q-OB, and antibodies towards A-beta needs to be performed. What types of aggregated pathologies, such as CAA, Cored and diffused A-beta plaques, as well as tau neurofibrillary tangles, are stained by Q-OB in human AD tissue sections? Such experiments are essential to assess the ligand's selectivity towards amyloid-beta oligomers from a clinical perspective.

Answer to referees:

Reviewer #1:

I genuinely appreciate the authors' commendable efforts in addressing my queries through rigorous experimentation. Their commitment has undoubtedly resulted in a more robust and captivating manuscript.

Response: We sincerely appreciate your positive comments to agree with accepting the current manuscript. Following your valuable feedback, we have successfully improved overall quality of the manuscript and are inspired to consider our next study plans to expand diagnostic platforms for advanced clinical applications toward Alzheimer's disease.

Reviewer #2:

Comments on manuscript NCOMMS-23-12055 submitted to the Nature Communications (Round 2)

TITLE: "Pioneering the Early Diagnosis of Alzheimer's Disease: A β Oligomers-Sensing Probe for Cerebrospinal Fluid Analysis in Patients"

Authors: Jusung An, Kyeonghwan Kim, Ho Jae Lim, Jinwoo Shin, In Wook Park, Illhwan Cho, Hyeong Yun Kim, Sunghoon Kim, YoungSoo Kim, Kun Ho Lee and Jong Seung Kim

In my opinion the answers given by the authors to the questions raised are convincing and the changes introduced as requested.

Therefore, I believe that the article deserves publication in Nature Communications.

Response: We would like to express our gratitude for your positive agreement to accept the current manuscript. Now, we strongly believe that your valuable comments have significantly improved the manuscript.

Reviewer #4:

The paper by Kim and co-workers presents the synthesis of a novel ligand for fluorescent detection of one of the pathological entities, amyloid-beta oligomers, seen in Alzheimer's disease.

The design and synthetic routes for generating the ligand, Q-OB, as well as the photophysical characteristics are well-presented. In addition, the ligands selectivity towards amyloid-beta oligomers is well characterized in in vitro fibrillation assays with recombinant A-beta species and the authors are also investigating the ligands performance for optical in vivo imaging in transgenic mouse models, as well as for detection of amyloid-beta oligomers in cerebrospinal fluid samples from patients.

However, to conclude that the performance of Q-OB is restricted to detection of amyloid-beta oligomers, as well as suitable for optical in vivo imaging, further experiments are essential.

In general, the specificity of the dyes towards amyloid-beta oligomers needs to be verified in tissue

samples. I would therefore recommend this paper for publication in Nature Communications after the revisions outlined below.

Query 1. The characterization of the ligand, Q-OB, for in vivo imaging, is not optimal. The authors only show macroscopic images (Figure 4 and Supplementary Figures 33 and 34) and images showing the characterization of Q-OB towards different aggregated A-beta species in tissue sections would be essential. In this regard, I suggested that histological staining of brain tissue sections from 5xFAD transgenic mice (young and old) and B6 wild type controls with the ligand, Q-OB, and antibodies towards A-beta should be performed. Is Q-OB staining the ordinary aggregated A-beta pathologies, such as CAA and parenchymal plaques? Is it possible to see oligomeric A-beta? Is there any unspecific staining of the white matter? A figure with zoom in images of the respective pathological entities would be preferable.

Response: We greatly appreciate your informative review to raise the quality of our manuscript. As per your comments, we conducted an optical characterization of **Q-OB** towards amyloid pathologies in brain tissue sections. The results of histological staining demonstrated that **Q-OB** dye can stain amyloid- β ($A\beta$) aggregates without unspecific staining of the white matter in comparison with a widely used anti- $A\beta$ monoclonal antibody 6E10 in brain tissues of 5xFAD transgenic mice and B6 wild-type controls. **Q-OB**-treated brain tissue slice isolated from 12-months-old 5xFAD mice showed different staining levels in various concentration ranges from 0.01 to 1 μ M, and 1 μ M of **Q-OB** highly distinguished $A\beta$ aggregates at red and green emission ($\lambda_{ex}/\lambda_{em} = 520/590$ and $\lambda_{ex}/\lambda_{em} = 480/520$ nm, respectively) as shown in **Figure 1** below, added to the revised supplementary information as Figure S29.

Figure 1. Fluorescence images of cortex area of brain tissue slice from 12-months-old 5xFAD transgenic Alzheimer's disease (AD) model mice stained by various concentrations of **Q-OB** (0.01, 0.1, 0.5, and 1 μ M) (red channel; $\lambda_{em}/\lambda_{ex} = 520/590$ nm and green channel; $\lambda_{em}/\lambda_{ex} = 480/520$ nm). Scale bars indicate the length stated in the figure.

Interestingly, we unexpectedly discovered that the fluorescent signal of **Q-OB** towards A β aggregates simultaneously demonstrated differentiated staining patterns between green and red channels. The red channel exhibited that **Q-OB** dye stains dense core (compact) plaques as a small dot shape. In contrast, diffuse plaque-like patterns were observed at the green channel (**Figure 2** below, added to the revised supplementary information as Figure S30). In addition, **Q-OB**-stained amyloid plaques were exhibited in a brain tissue slice from 16-months-old 5xFAD mice compared to the histological staining using 6E10 (1:200, Biolegend, USA, #SIG-39320), as a primary antibody and anti-mouse IgG-Alexa 350 (1:200, Invitrogen, USA, #A11045) as a secondary antibody as depicted in **Figure 3** below (Figure 4a and S31 in the revised manuscript and the supplementary information). Merge of **Q-OB** (red and green channels) and 6E10 (blue channel) signals showed three different staining patterns as magenta (overlap of red and blue channels), red (red channel without overlapping), cyan (overlapping green and blue channels), and white (overlapping red, green, and blue channels) pseudo-colors as shown in **Figure 4** below (partially added to the revised manuscript as Figure 4b and the revised supplementary information as Figure S32). For reference, the sporadic tiny dot-shape fluorescence signal at the red channel was caused by tissue autofluorescence. It was directly proved by control mouse brain tissue image of B6 wild-type mouse with/without **Q-OB** and 5xFAD transgenic AD model mouse with staining of **Q-OB** (**Figure 5** below, Figure S33 in the revised supplementary information).

These results indicated that **Q-OB** dye may visualize different types of A β aggregates at the same time, and it can be inferred that fluorescence signals for A β aggregates other than typical A β plaques are generated, especially considering that a staining pattern that is distinct from those of 6E10.

Figure 2. a) Fluorescence images of cortex area of the brain tissue slice of 12-months-old 5xFAD transgenic AD model mouse stained by **Q-OB** (1 μM) (red channel; $\lambda_{em}/\lambda_{ex} = 520/590 \text{ nm}$ and green channel; $\lambda_{em}/\lambda_{ex} = 480/520 \text{ nm}$). b) Magnified images of aggregated amyloid- β (A β) pathologies at the white dot-line boxed area. Scale bars indicate the length stated in the figure.

Figure 3. Fluorescence images of cortex area of the brain tissue slice (16-months-old 5xFAD transgenic AD model mouse) stained by **Q-OB** (1 μ M, red and green channels) and 6E10 (anti-A β antibody, 1:200, blue channel) (red channel; $\lambda_{em}/\lambda_{ex} = 520/590$ nm, green channel; $\lambda_{em}/\lambda_{ex} = 480/520$ nm, and blue channel; $\lambda_{em}/\lambda_{ex} = 360/470$ nm) ($n = 3$). Scale bars indicate the length stated in the figure.

Figure 4. Fluorescence images of cortex area of the brain tissue slice from 16-months-old 5xFAD transgenic AD model mouse stained by **Q-OB** (1 μ M, red and green channels) and IHC using 6E10 (anti-A β antibody, 1:200, blue channel) (red channel; $\lambda_{em}/\lambda_{ex}$ = 520/590 nm, green channel; $\lambda_{em}/\lambda_{ex}$ = 480/520 nm, and blue channel; $\lambda_{em}/\lambda_{ex}$ = 360/470 nm). Magnified images of A β aggregates at the white dot-line boxed area for (i) and (ii), respectively. i) Stained A β aggregates are indicated by hollowed arrow (an overlap of red and blue channels), and white arrow (red channel without overlapping). ii) Stained A β aggregates are marked by hollowed arrow (overlapping of red, green, and blue channels), and white arrow (an overlap of green and blue channels). Yellow arrows in Zoom-in images demonstrate autofluorescence. Scale bars indicate the length stated in the figure.

Figure 5. Fluorescence images of cortex area of the brain tissue isolated from 6-months-old B6 wild-type mouse and 16.5-months-old 5xFAD transgenic AD model mouse with or without staining using **Q-OB** (1 μ M, red and green channels) and IHC using 6E10 (anti-A β antibody, 1:200, blue channel) (red channel; $\lambda_{em}/\lambda_{ex}$ = 520/590 nm, green channel; $\lambda_{em}/\lambda_{ex}$ = 480/520 nm, and blue channel; $\lambda_{em}/\lambda_{ex}$ = 360/470 nm). White dot-line boxes indicate magnified area for a, b, and c, respectively, and yellow arrows in Zoom-in images demonstrate autofluorescence. Scale bars indicate the length stated in the figure.

To investigate different fluorescence responses of **Q-OB** observed at red and green channels, we conducted a time-point incubation assay using **Q-OB** in various conditions, including brain lysates of 5xFAD transgenic mouse, B6 wild-type mouse, phosphate-buffered saline (PBS) containing 10 μ M of A β_{1-42} (10 mM, pH = 7.4), radioimmunoprecipitation assay (RIPA) buffer (pH = 8.0), and deionized water (DW) at various time points (0, 2, 3, 12, and 24 h) (**Figure 6** below). Unexpectedly, a significant hypochromic shift (approximately 60 nm) was observed, and its intensities were gradually enhanced for 24 h incubation in brain lysates of both AD transgenic and wild-type mice. In contrast, negligible spectral changes were found in buffer solution and DW. Consequently, we believe that non-specific chemical or physical interactions may occur in brain physiological environments, which induce dramatic changes in the photophysical properties of **Q-OB** probe. We have added the data and discussions to the revised manuscript and the supplementary information.

Figure 6. Fluorescence spectroscopy of **Q-OB** for identifying $A\beta_{1-42}$ species (10 μM) at various time points (0, 2, 3, 12, and 24 h) of incubation in (a) brain lysate of 5xFAD transgenic mouse (Tg), (b) brain lysate of B6 wild-type mouse (WT), (c) PBS (10 mM, pH = 7.4), (d) radioimmunoprecipitation assay (RIPA) buffer (pH = 8.0), and (e) deionized water (DW). f) Fluorescence intensity changes of **Q-OB** at 540 nm from data (a–e).

By the way, as thoroughly discussed in the first revision phase, the reason why we did not optimize characterization of **Q-OB** for optical bioimaging is that we would like to emphasize an *in vitro* kinetic assay for early-AD diagnosis using patients' biofluid as a primary potential application. To the best of our knowledge, current staining agents, including small-molecule fluorophore or anti- $A\beta$ antibodies, are unsuitable for specific visualizing $A\beta$ oligomers in the cerebral cortical tissue without interference of amyloid plaques. Nevertheless, some previous works claim the unconvincing staining potential of developed dyes to $A\beta$ oligomers through optical imaging results (C. L. Teoh *et al.*, Chemical Fluorescent Probe for Detection of $A\beta$ Oligomers, *J. Am. Chem. Soc.* 137, 13503–13509, DOI: 10.1021/jacs.5b06190 (2015); Y. Li *et al.*, Fluoro-substituted cyanine for reliable *in vivo* labelling of amyloid- β oligomers and neuroprotection against amyloid- β induced toxicity *Chem. Sci.*, 8, 8279–8284, DOI: 10.1039/C7SC03974C (2017); L. Sun *et al.*, Amphiphilic Distyrylbenzene Derivatives as Potential Therapeutic and Imaging Agents for Soluble and Insoluble Amyloid β Aggregates in Alzheimer's Disease, *J. Am. Chem. Soc.* 143, 10462–10476, DOI: 10.1021/jacs.1c05470 (2021)).

Regarding our finding, it is still shrouded in mystery to prove whether the stained analytes in the brain tissue section were actually $A\beta$ oligomers. Indeed, anti- $A\beta$ antibodies 6E10 and A11 could mainly stain amyloid plaques in AD brain, and in most cases, even small-molecule-based $A\beta$ oligomer dyes also mark the amyloid plaque region. Since amyloid plaque area in the brain tissue, unlike the artificially formed $A\beta$ solution, is significantly heterogeneous in the composition of various $A\beta$ species, which acts as a crucial hurdle in selectively visualizing only certain $A\beta$ states, such as $A\beta$ oligomers—transiently exist in the physiological condition. In other words, in order to assert that $A\beta$ oligomers are specifically stained in the brain tissue section, additional validations are required to explain authenticity of whether the fluorescent signal is induced by the stained target

oligomers. Therefore, we conclude that it is unable to specifically label A β oligomers in the brain tissue using developed fluorescent probes in the current work. Nonetheless, we appreciate your suggestion for discovering an interesting phenomenon regarding **Q-OB** dye and plan to design a novel staining method that can visually analyze A β oligomers in brain tissue for our future research. We strongly believe the fact that the typical staining pattern of 6E10 is significantly different from that of the **Q-OB** dye suggests the potential development of an A β oligomer staining method in the AD brain tissue in the future.

Query 2. Since a variety of studies, including high resolution structures obtained by cryo electron microscopy, have shown that the structure of *in vitro* generated aggregated A-beta species, as well as mouse derived A-beta aggregates, are different from aggregated A-beta pathology derived from human AD tissue sections, histological staining of brain tissue sections from human AD cases and age matched controls with the ligand, Q-OB, and antibodies towards A-beta needs to be performed. What types of aggregated pathologies, such as CAA, Cored and diffused A-beta plaques, as well as tau neuro fibrillary tangles, are stained by Q-OB in human AD tissue sections? Such experiments are essential to assesses the ligands selectivity towards amyloid-beta oligomers from a clinical perspective.

Response: Thanks for your valuable feedback on this clinically important point. As per your mentioned, physicochemical and biological properties of aggregated A β pathology derived from human AD brain tissue significantly vary artificially generated A β species *in vitro* and/or transgenic AD model mice. Thus, encouraged by mice imaging findings, we investigated labeling capability of **Q-OB** to A β aggregates in the brain tissue of human AD patients. For this, we carefully collected postmortem cerebral hippocampal tissue specimens from Victorian Brain Bank Network (VBBN), supported by The Florey, The Alfred, and the Victorian Institute of Forensic Medicine (**Table 1** below, added to the revised supplementary information as Table S4). The cerebral hippocampal tissue specimens were obtained from a total of 13 individuals (AD patient group of four $\epsilon 3$ homozygotes, two $\epsilon 2/\epsilon 3$ individuals, and one $\epsilon 3/\epsilon 4$ individual and non-patient group of five $\epsilon 3$ homozygotes and one $\epsilon 2/\epsilon 3$ individual) and controls are all individuals who have a full neuropathological examination and have no significant neuropathology.

Formalin-fixed and paraffin-embedded human postmortem cerebral hippocampal tissue were immersed in xylene three times for 10 min to remove paraffin, followed by two 10 min washes with a series of diluted alcohols (100%, 90%, 70%, and 50% EtOH in DW). After washing, antigen was retrieved with citrate buffer (10 mM, pH = 6.0) at 95°C for 30 min. The brain slices were moved to the refrigerator for 1 h and stained with **Q-OB** and anti-A β antibodies (6E10 or 4G8) under the same conditions as the mice brain slices.

The results of histological staining using 6E10 exhibited that **Q-OB** dye could mark extracellular A β aggregates, especially senile (amyloid) plaques, in the hippocampus area of brain tissue from AD-diagnosed patients (**Figure 7** below, added to the revised manuscript and supplementary information as Figure 4c and Figure S34). Similar to the mice brain staining, sporadic tiny dot-shape fluorescence signal was observed at the red channel induced by tissue autofluorescence. Negligible non-specific binding was observed on the cognitive normal controls with staining of **Q-OB** (**Figure 8** below, Figure S35 in the revised supplementary information).

On the other hand, unlike mice brain tissue, **Q-OB** staining images acquired at red and green channels showed different staining patterns. At green channel, **Q-OB**-stained senile plaques were

widely merged with 6E10 (most of the senile plaques), and red channel images were partially merged with 6E10 nearby plaques (**Figure 9** below, Figure S36 in the revised supplementary information). Consequently, the merge of **Q-OB** (red/green channels) and 6E10 (blue channel) response demonstrated various staining patterns, including yellow (overlap of red and green channels), cyan (overlapping of green and blue channels), and white (overlap of red, green, and blue channels) pseudo-colors as shown in **Figure 10** below (Figure S37 in the revised supplementary information and partially added to the revised manuscript as Figure 4d). Taken together, these observations indicate that **Q-OB** dye can stain senile plaques at green channel merged with those of 6E10, not red.

Table 1. Information of postmortem cerebral hippocampal tissue specimens.

No	Diagnosis	Age	Sex	PMI (h)	APOE	Staining methods
1	ADD	54.2	M	49.5	$\epsilon 3/\epsilon 3$	
2	ADD	70.3	M	13.5	$\epsilon 3/\epsilon 3$	
3	ADD	73.3	F	7.5	$\epsilon 3/\epsilon 4$	
4	ADD	80.3	F	47	$\epsilon 2/\epsilon 3$	
5	ADD	80.6	M	24	$\epsilon 3/\epsilon 3$	Q-OB/6E10 co-staining
6	CN	72.6	M	42.5	$\epsilon 3/\epsilon 3$	
7	CN	73.5	M	22	$\epsilon 3/\epsilon 3$	
8	CN	78.3	M	46	$\epsilon 3/\epsilon 3$	
9	CN	78.8	F	19	$\epsilon 2/\epsilon 3$	
10	CN	79.3	M	57	$\epsilon 3/\epsilon 3$	
11	ADD	63.5	M	43	$\epsilon 2/\epsilon 3$	
12	ADD	83	F	34	$\epsilon 3/\epsilon 3$	Q-OB/4G8 co-staining
13	CN	59	F	30	$\epsilon 3/\epsilon 3$	

Written informed consent was obtained from each participant or their legal guardian. Diagnostic information was designated by the Victorian Brain Bank Network (VBBN), supported by The Florey, The Alfred, and the Victorian Institute of Forensic Medicine. The abbreviated words are Alzheimer’s disease dementia (ADD), Cognitive normal (CN), Female (F), Male (M), post-mortem interval (PMI), and apolipoprotein E (APOE).

Figure 7. Fluorescence images of hippocampus area of the brain tissue slice (AD-diagnosed patients) stained by **Q-OB** (1 μM , red and green channels) and **6E10** (anti-A β antibody, 1:200, blue channel) (red channel; $\lambda_{em}/\lambda_{ex} = 520/590 \text{ nm}$, green channel; $\lambda_{em}/\lambda_{ex} = 480/520 \text{ nm}$, and blue channel; $\lambda_{em}/\lambda_{ex} = 360/470 \text{ nm}$) ($n = 5$). Scale bars indicate the length stated in the figure.

Figure 8. Fluorescence images of hippocampus area of the brain tissue slice of healthy human (non-AD-patient) stained by **Q-OB** (1 μ M, red and green channel) and 6E10 (anti-A β antibody, 1:200, blue channel) (red channel; $\lambda_{em}/\lambda_{ex} = 520/590 \text{ nm}$, green channel; $\lambda_{em}/\lambda_{ex} = 480/520 \text{ nm}$, and blue channel; $\lambda_{em}/\lambda_{ex} = 360/470 \text{ nm}$) ($n = 5$). Scale bars indicate the length stated in the figure.

Figure 9. Fluorescence images of hippocampus area of the brain tissue slice from human AD patients stained by Q-OB (1 μ M, red and green channels) and IHC using 6E10 (anti-A β antibody, 1:200, blue channel) (red channel; $\lambda_{em}/\lambda_{ex}$ = 520/590 nm, green channel; $\lambda_{em}/\lambda_{ex}$ = 480/520 nm, and blue channel; $\lambda_{em}/\lambda_{ex}$ = 360/470 nm). Scale bars indicate the length stated in the figure.

Figure 10. Fluorescence images of hippocampus area of the brain tissue slice collected from human AD patients. a) Magnified images of A β aggregates stained by Q-OB (1 μ M, red and green channels) and 6E10 (anti-A β antibody, 1:200, blue channel) (red channel; $\lambda_{em}/\lambda_{ex}$ = 520/590 nm, green channel; $\lambda_{em}/\lambda_{ex}$ = 480/520

nm, and blue channel; $\lambda_{em}/\lambda_{ex} = 360/470$ nm). b) Stained A β aggregates are indicated by hollowed arrow (an overlap of green and blue channels), white arrow (an overlap of red, green and blue channels), and yellow arrow (an overlap of red and green channels). Scale bars show the length stated in the figure.

Quantification of the number and pseudo-color of fluorescence-detected spots demonstrates the labeling capacity of **Q-OB** to A β aggregates compared to immunostaining (**Figure 11** below). The primary color indicated that the A β aggregates stained by **Q-OB** (red or green) or 6E10 (blue) do not overlap each other, and the secondary colors are the overlap of two **Q-OB** staining patterns (yellow, overlap of red and green) or the case where **Q-OB** and 6E10 overlap (magenta, overlap of red and blue; cyan, overlapping green and blue). A β aggregates stained exclusively by **Q-OB** in mainly brain tissue of transgenic AD model mice as shown in primary color (red/green) and secondary color (yellow). Whereas, in human tissue specimens, senile plaques stained by **Q-OB** were observed to overlap with immunostaining mostly as can be seen from the spots marked with magenta and cyan colors.

Figure 11. Quantification of the number and pseudo-color of fluorescence-stained A β aggregated spots with **Q-OB** or 6E10. Secondary color indicates an overlap of two primary color (yellow, red and green channels; magenta, red and blue channels; cyan, green and blue channels) and tertiary color indicates overlapping three primary colors (white).

In addition, cerebral amyloid angiopathy was visualized with **Q-OB** dye as a hollowed ring-shaped accumulation of A β within the cerebral blood vessels in a green channel well-merged with 6E10(blue channel), as can be seen from **Figure 12** below (added to the revised supplementary information as Figure S38). Furthermore, additional fluorescence optical images were obtained using 4G8 anti-A β antibody (1:200, Biologend, USA, #800701), which recognizes residues 18–23 of the A β sequence. Similar to those of 6E10, 4G8 exhibited stained senile plaques overlapped with green channel of **Q-OB** fluorescence response (**Figure 13** below, added to the revised supplementary information as Figure S39).

Figure 12. Fluorescence images of hippocampus area of the brain tissue slice from human AD patients stained by Q-OB (1 μ M, red and green channels) and 6E10 (anti-A β antibody, 1:200, blue channel) (red channel; $\lambda_{em}/\lambda_{ex} = 520/590$ nm, green channel; $\lambda_{em}/\lambda_{ex} = 480/520$ nm, and blue channel; $\lambda_{em}/\lambda_{ex} = 360/470$ nm). Hollowed arrows demonstrate vascular A β . Scale bars indicate the length stated in the figure.

Figure 13. Fluorescence images of hippocampus area of the brain tissue slice of healthy human (non-AD-patient) and human AD patients stained by **Q-OB** (1 μ M, red and green channels) and 4G8 (anti-A β antibody, 1:200, blue channel) (red channel; $\lambda_{em}/\lambda_{ex}$ = 520/590 nm, green channel; $\lambda_{em}/\lambda_{ex}$ = 480/520 nm, and blue channel; $\lambda_{em}/\lambda_{ex}$ = 360/470 nm) (n = 3). Scale bars indicate the length stated in the figure.

In conclusion, it is currently challenging to specifically visualize A β oligomers using small-molecule-based dyes developed so far in the brain tissue of transgenic AD model mice as well as human AD patients without the interference of amyloid plaques. Nevertheless, what is still interesting is that the staining pattern of **Q-OB** partially varies from those of 6E10 or 4G8 typical anti-A β antibodies, which specifically stain amyloid plaques in the brain tissue of AD patients and transgenic mice. Even so, it is currently debatable to argue that only A β oligomers are selectively stained by **Q-OB** in our findings; even **Q-OB** has a significant selectivity and specificity for A β oligomers *in vitro*. Hence, utilization of **Q-OB** is currently unconvincing for A β oligomers-specific optical imaging in the brain region as a clinical application. Indeed, powerful imaging agents are able to visualize A β oligomers in brain tissue and *in vivo* models are believed to be necessary for a detailed understanding of A β -related pathology, disease diagnosis, and effective treatment monitoring of AD. In this respect, **Q-OB** provides a great possibility for further development of advanced probes applicable in the fundamental and preclinical studies of AD associated with A β oligomer sensing. Therefore, our next study plans involve utilizing **Q-OB** analogues as an ideal starting point for developing a novel A β oligomer-specific visualizing tool, which has never been available before, through revealing the fundamental reason for the occurrence of different staining patterns between **Q-OB** dye and immunostaining using anti-A β antibodies.

Reviewers' Comments:

Reviewer #4:

Remarks to the Author:

The authors' have addressed most of my previous concerns, including additional experiments and evaluation of the Q-OB ligand on tissue sections from transgenic mice and human AD cases. However, there is some remaining issues regarding the interpretation of the results that needs to be clarified before I would recommend this paper for publication in Nature Communications. Particularly, the novel data obtained from the tissue staining experiments undoubtedly show that the performance of Q-OB is not restricted to detection of amyloid-beta oligomers, so the manuscript needs to be revised accordingly.

1. When examining the results from staining of human tissue sections, it is evident that the ligand, Q-OB, stains the classical fibrillar A-beta pathologies, such as CAA, dense core and diffuse parenchymal plaques. Thus, in human AD brain tissue samples, Q-OB is clearly not selective/specific for oligomeric A-beta ($\alpha\text{A}\beta$), since the fibrillar pathologies are also detected by the ligand. Therefore, the authors need to avoid using the term $\alpha\text{A}\beta$ selective/specific ligand throughout the manuscript. The correct wording would be, as stated in the title, $\text{A}\beta$ Oligomers-Sensing Probe, and this matter needs to be corrected in several parts of the manuscript. In addition, the schematic illustration in figure 1 needs to be revised since the ligands also stains amyloid fibrils in the AD brain. At the moment, the illustration only shows that Q-OB are selective for prefibrillar A-beta oligomers and the experiments on human AD tissue sections clearly confirms that this is not the case.

2. In the results/discussion sections, line 360 the authors state:

"Merge of Q-OB (red and green channels) and 6E10 (blue channel) pseudo-colored signal showed that three different staining patterns as magenta (overlap of red and blue channels), red (red channel without overlapping), cyan (overlapping green and blue channels), and white (overlap of red, green, and blue channels) colors induced by the different overlap of each channel (Figures 4b and S30–33). These results indicated that Q-OB dye may visualize different types of aggregated $\text{A}\beta$ pathologies at the same time, and it can be inferred that fluorescence signals for $\text{A}\beta$ aggregates other than typical $\text{A}\beta$ plaques are generated, especially considering that a staining pattern that is distinct from those of 6E10."

The last sentence regarding that the staining pattern of Q-OB is different compared to the 6E10 antibody staining is highly speculative, since there is now evidence that the fluorescent signal lacking the overlap with 6E10 staining is deposits composed of A-beta. In fact, as shown in Figure S32 there is a lot of auto-fluorescent structures (high-lighted with yellow arrows), such as lipofuscin, observed in the green/red channel. Therefore, this sentence should be removed. In addition, the figure legend of figure 4 should be updated, since it is lacking the explanation for the auto-fluorescent structures that are indicated with yellow arrows. Overall, the authors should check most of the figure legends, both in the main manuscript and the supporting information, since there are several discrepancies between labeling in the figures and the legends.

3. The interpretation of the results from the histological staining of the human AD tissue sections needs some major revision and some additional co-staining experiments with antibodies, such as AT8 or AT100, targeting aggregated tau pathologies are essential.

- Firstly, when looking at figure 4D, Figure S36 and Figure S37, it is rather evident that some structures are only observed with Q-OB fluorescence (green/red channel). Morphological, although the images are collected with low magnification, these structures resemble aggregated tau-pathologies. Hence, Q-OB is not only staining A-beta aggregates, but the ligand is also most likely staining tau deposits, the second pathological hallmark in AD. The structures highlighted white yellow arrows (overlap of red and green channels) resembles neurofibrillary tangles and some A-beta deposits (low right corner in figure 4D) are clearly neuritic plaques with dystrophic neurites shown in yellow.

Therefore, the authors need to perform co-staining experiments with Q-OB and a tau specific antibody. My guess is that such experiments will show that the Q-OB also stains some aggregated tau pathologies in human AD brain.

- Secondly, the small granular specimens that show overlapping blue/green/red fluorescence that are highlighted with white dot-line circle in figure 4D and white arrow heads in figure S37b clearly resembles autofluorescent lipofuscin that are a major component in human brain tissue sections. These structures can also be observed in the control brains (figure S35) and the authors should discussed this issue in the text.

Answer to referee:

Reviewer #4:

The authors' have addressed most of my previous concerns, including additional experiments and evaluation of the Q-OB ligand on tissue sections from transgenic mice and human AD cases. However, there is some remaining issues regarding the interpretation of the results that needs to be clarified before I would recommend this paper for publication in Nature Communications. Particularly, the novel data obtained from the tissue staining experiments undoubtedly show that the performance of Q-OB is not restricted to detection of amyloid-beta oligomers, so the manuscript needs to be revised accordingly.

Query 1. When examining the results from staining of human tissue sections, it is evident that the ligand, Q-OB, stains the classical fibrillar A-beta pathologies, such as CAA, dense core and diffuse parenchymal plaques. Thus, in human AD brain tissue samples, Q-OB is clearly not selective/specific for oligomeric A-beta ($\text{oA}\beta$), since the fibrillar pathologies are also detected by the ligand. Therefore, the authors need to avoid using the term $\text{oA}\beta$ selective/specific ligand throughout the manuscript. The correct wording would be, as stated in the title, $\text{A}\beta$ Oligomers-Sensing Probe, and this matter needs to be corrected in several parts of the manuscript. In addition, the schematic illustration in figure 1 needs to be revised since the ligands also stains amyloid fibrils in the AD brain. At the moment, the illustration only shows that Q-OB are selective for prefibrillar A-beta oligomers and the experiments on human AD tissue sections clearly confirms that this is not the case.

Response: We appreciate your thorough review, which has undoubtedly enhanced the quality of our manuscript. We acknowledge your concern regarding the wording “selective (and/or specific) probe for amyloid- β ($\text{A}\beta$) oligomers” used in the manuscript after the histological staining results. Indeed, the **Q-OB** probe does not exclusively stain $\text{A}\beta$ oligomers, as fibrillar $\text{A}\beta$ aggregates are also detected.

However, it is important to note that this observation is limited to the tissue imaging. In controlled artificial conditions, such as solution tests or biofluid assays, which are the primary focus of our study, the concentration of $\text{A}\beta$ species, including $\text{A}\beta$ oligomers, can be kinetically regulated. In these conditions, we have characterized the **Q-OB** probe as ‘selective’ or ‘specific’ for each proteinaceous analyte, based on distinct photophysical and physicochemical properties such as fluorescent quantum yield, binding affinity, limit of detection, and more. This characterization is reasonable, especially in the context of solution-based experiments.

On the other hand, in a natural context where intrinsic concentrations of target analytes (i.e., $\text{A}\beta$ species) vary significantly due to the opposing physiological properties of transiently existing $\text{A}\beta$ oligomers and perpetually deposited fibrils. It becomes challenging to explain the selectivity or specificity of the **Q-OB** probe due to the extreme concentration difference. This significant difference in analyte concentration highlights the difficulty in developing a method capable of selectively and specifically visualizing $\text{A}\beta$ oligomers in brain tissue.

In response to your insightful comments, we have carefully revised and clarified the manuscript, particularly in terms of terminology and the schematic illustration, Figure 1 in the main text. We now provide a more nuanced description of the **Q-OB** probe’s features related to $\text{A}\beta$ oligomers, consistently referring to it as an “ $\text{A}\beta$ oligomer-sensing probe” to accurately represent

its staining capabilities. As previously mentioned in our revisions, while acknowledging the limitations of optical imaging using the **Q-OB** probe as the primary purpose, we emphasize the significance of our *in vitro* kinetic assay for early AD diagnosis, particularly in analyzing patients' cerebrospinal fluids.

Query 2. In the results/discussion sections, line 360 the authors state:

"Merge of Q-OB (red and green channels) and 6E10 (blue channel) pseudo-colored signal showed that three different staining patterns as magenta (overlap of red and blue channels), red (red channel without overlapping), cyan (overlapping green and blue channels), and white (overlap of red, green, and blue channels) colors induced by the different overlap of each channel (Figures 4b and S30–33). These results indicated that **Q-OB** dye may visualize different types of aggregated A β pathologies at the same time, and it can be inferred that fluorescence signals for A β aggregates other than typical A β plaques are generated, especially considering that a staining pattern that is distinct from those of 6E10."

The last sentence regarding that the staining pattern of Q-OB is different compared to the 6E10 antibody staining is highly speculative, since there is no evidence that the fluorescent signal lacking the overlap with 6E10 staining is deposits composed of A-beta. In fact, as shown in Figure S32 there is a lot of auto-fluorescent structures (high-lighted with yellow arrows), such as lipofuscin, observed in the green/red channel. Therefore, this sentence should be removed. In addition, the figure legend of figure 4 should be updated, since it is lacking the explanation for the auto-fluorescent structures that are indicated with yellow arrows. Overall, the authors should check most of the figure legends, both in the main manuscript and the supporting information, since there are several discrepancies between labeling in the figures and the legends.

Response: We sincerely appreciate your valuable comments in improving the precision and clarity of our interpretation of the results. Regarding the statement about the staining pattern of **Q-OB** being different from the 6E10 antibody staining, we understand your point about the speculative nature of this assertion, especially considering the evidence presented in Figure S32. Furthermore, we acknowledge that the presence of auto-fluorescent structures, such as lipofuscin in the green/red channel, could contribute to the observed differences in fluorescence signals. In response to this, we have revised the paragraph to describe the fluorescent signal lacking overlap with 6E10 staining as deposits composed of A β , removing the speculative statements, and providing a more explicit interpretation of the results in that context.

Additionally, we appreciate your attention to detail regarding Figure 4 in the main text and the suggestion to update the figure legend to explain the auto-fluorescent structures indicated with yellow arrows. We have revised the figure legend to provide an accurate description of these structures in the revised manuscript and the supplementary information. Furthermore, we have conducted a thorough review of all figure legends to address any discrepancies and ensure accurate descriptions.

Query 3. The interpretation of the results from the histological staining of the human AD tissue sections needs some major revision and some additional co-staining experiments with antibodies, such as AT8 or AT100, targeting aggregated tau pathologies are essential.

- Firstly, when looking at figure 4D, Figure S36 and Figure S37, it is rather evident that some structures are only observed with Q-OB fluorescence (green/red channel). Morphological, although the images are collected with low magnification, these structures resemble aggregated tau-pathologies. Hence, Q-OB is not only staining A β aggregates, but the ligand is also most likely staining tau deposits, the second pathological hallmark in AD. The structures highlighted with white yellow arrows (overlap of red and green channels) resembles neurofibrillary tangles and some A-beta deposits (low right corner in figure 4D) are clearly neuritic plaques with dystrophic neurites shown in yellow. Therefore, the authors need to perform co-staining experiments with Q-OB and a tau specific antibody. My guess is that such experiments will show that the Q-OB also stains some aggregated tau pathologies in human AD brain.

- Secondly, the small granular specimens that show overlapping blue/green/red fluorescence that are highlighted with white dot-line circle in figure 4D and white arrow heads in figure S37b clearly resembles autofluorescent lipofuscin that are a major component in human brain tissue sections. These structures can also be observed in the control brains (figure S35) and the authors should discuss this issue in the text.

Response: Thank you for your thoughtful evaluation of our histological staining results and for providing specific suggestions for improvement. Your detailed feedback greatly helps refine the precision and rigor of our study. Based on your comments, we have thoroughly implemented the suggested phosphorylated tau (p-Tau) co-staining experiments and included a discussion addressing the presence of autofluorescent lipofuscin in whole brain sections.

Aggregated Tau Pathologies: We acknowledge your observation that certain structures observed in the **Q-OB** fluorescence (specifically within the green/red channels) may not resemble aggregated tau pathologies, as seen in **Figure 1** below. We conducted co-staining experiments using **Q-OB** (1 μ M) dye and the anti-p-Tau antibody, AT8 (pS202/pT205-tau, Invitrogen, #MN1020, 1:200), as the primary antibody. We used anti-mouse IgG (Alexa-350 conjugated, blue, 1:200) as the secondary antibody applied to hippocampal tissues from AD-patients. AT8 demonstrated distinct staining of p-Tau aggregates (yellow circle), which did not properly merge with **Q-OB**. However, due to the strong fluorescence intensity from **Q-OB** and the low responsiveness of AT8-stained p-Tau, several spotted patterns of **Q-OB** were observable in each channel, especially in the blue channel, which appeared white in the merged image, indicated by the yellow arrowheads. Additionally, the interference caused by autofluorescent factors hindered the accurate identification of tau-labeling by **Q-OB**. Since **Q-OB**-labeled deposits (mainly A β accumulation) did not co-localize significantly with AT8-stained p-Tau aggregates, the detection capability of **Q-OB** dye for p-Tau is negligible.

Autofluorescent Lipofuscin: we appreciate your identification of small granular specimens resembling autofluorescent lipofuscin, highlighted with a white dot-line circle in Figure 4D and white arrowheads in Figure S37b. We have included a detailed discussion in the revised manuscript, acknowledging the presence of autofluorescent lipofuscin and its observation in control brains.

We hope these revisions address your concerns and contribute to the clarity and precision of our manuscript. Your guidance has been invaluable in refining the scientific rigor of our work.

Figure 1. Fluorescence images of hippocampal area of the brain tissues (AD-diagnosed patients) stained by **Q-OB** (1 μ M, red and green channels) and **A8T** (anti-phosphorylated tau antibody, 1:200, blue channel) ($n = 3$). Scale bars indicate the length state in the figure. Stained phosphorylated tau is indicated by yellow circle. Spotted pattern of **Q-OB** and autofluorescence were observed in each channel which colored white in merged image (yellow arrowheads).

Reviewers' Comments:

Reviewer #4:

Remarks to the Author:

The authors have address most of my previous concerns and therefore, I would recommend this manuscript for publication in Nature Communications.